# HSurf-Net: Normal Estimation for 3D Point Clouds by Learning Hyper Surfaces

**Qing Li**[1]    **Yu-Shen Liu**[1*]    **Jin-San Cheng**[2]    **Cheng Wang**[3]
**Yi Fang**[4]    **Zhizhong Han**[5]

[1]School of Software, BNRist, Tsinghua University, Beijing, China
[2]Academy of Mathematics and System Sciences, Chinese Academy of Sciences, Beijing, China
[3]School of Informatics, Xiamen University, Xiamen, China
[4]Center for Artificial Intelligence and Robotics, New York University Abu Dhabi, Abu Dhabi, UAE
[5]Department of Computer Science, Wayne State University, Detroit, USA
{leoqli, liuyushen}@tsinghua.edu.cn   jcheng@amss.ac.cn
cwang@xmu.edu.cn   yfang@nyu.edu   h312h@wayne.edu

## Abstract

We propose a novel normal estimation method called HSurf-Net, which can accurately predict normals from point clouds with noise and density variations. Previous methods focus on learning point weights to fit neighborhoods into a geometric surface approximated by a polynomial function with a predefined order, based on which normals are estimated. However, fitting surfaces explicitly from raw point clouds suffers from overfitting or underfitting issues caused by inappropriate polynomial orders and outliers, which significantly limits the performance of existing methods. To address these issues, we introduce hyper surface fitting to implicitly learn *hyper surfaces*, which are represented by multi-layer perceptron (MLP) layers that take point features as input and output surface patterns in a high dimensional feature space. We introduce a novel space transformation module, which consists of a sequence of local aggregation layers and global shift layers, to learn an optimal feature space, and a relative position encoding module to effectively convert point clouds into the learned feature space. Our model learns hyper surfaces from the noise-less features and directly predicts normal vectors. We jointly optimize the MLP weights and module parameters in a data-driven manner to make the model adaptively find the most suitable surface pattern for various points. Experimental results show that our HSurf-Net achieves the state-of-the-art performance on the synthetic shape dataset, the real-world indoor and outdoor scene datasets. The code, data and pretrained models are publicly available at https://github.com/LeoQLi/HSurf-Net.

## 1 Introduction

Estimating normals from point clouds is vital for various downstream applications of 3D computer vision, such as point cloud filtering [5, 46, 35, 34], surface reconstruction [27] and rendering [8, 15, 40]. Though this topic has been extensively studied, it is still a challenge to work on point clouds with different types of noise, outliers, and density variations. It is well-known that point cloud normal estimation can be formulated as the least squares optimization problem [12], which explicitly fits a geometric surface (e.g. plane and polynomial surface) on local neighboring points and then computes the normal from the fitted surface. Specifically, the point-wise weights dedicate the importance of each point for the surface fitting.

---

[*]Correspondig author

36th Conference on Neural Information Processing Systems (NeurIPS 2022).

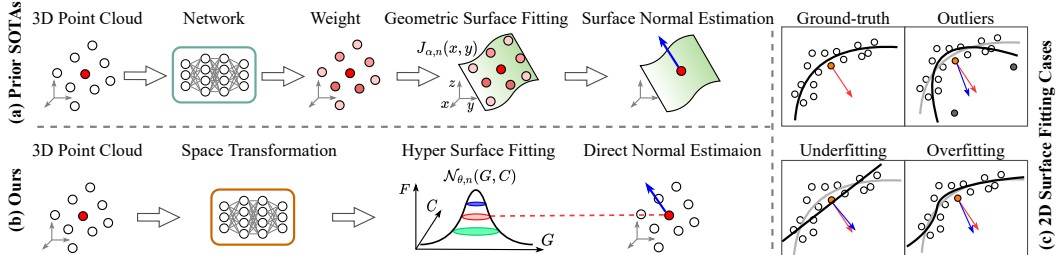

Figure 1: (a) Prior SOTA methods focus on weight learning and geometric surface fitting to estimate the surface normal. (b) We use features $G, C$ to learn a hyper surface $\mathcal{N}_{\theta,\tau}(G, C)$ to directly estimate normal for each point. (c) Existing surface fitting techniques are severely affected by overfitting, underfitting and outliers, which lead to inaccurate surface approximation and normal estimation.

Existing normal estimation algorithms can be roughly divided into two categories: traditional methods and learning-based methods. The traditional ones usually approximate potential structural properties of point clouds and use a well-designed algorithm to fit local planes or polynomial surfaces [21, 12]. However, explicitly fitting geometric surfaces heavily rely on careful parameters tuning, such as the neighborhood size and the order of polynomial function. The learning-based ones initially try to directly predict normal vectors from point clouds through regression network [10, 17, 7, 53, 52]. Alternatively, recent methods employ Convolutional Neural Networks (CNN) [30] or the PointNet architecture [6, 55] to learn point-wise weights, then the classic geometric surface fitting is utilized to compute normals (see Fig. 1(a)). The learning-based regression methods lack geometric prior and are hard to learn a general mapping from severely degraded inputs to the ground-truth normals, especially for complex geometric structures. For the learning-based surface fitting methods, one issue of these approaches is that explicit surface fitting is sensitive to noise and outliers. The weights on noisy points that are far away from the underlying surface significantly affect the accuracy of normals, even small weights on outliers still result in an erroneous normal estimation. Another inherent issue is that their predefined polynomial functions may not be suitable to fit various surfaces since a constant order of polynomial functions is selected for all points, e.g. plane in [30, 11] and 3-jet surface in [6, 55]. If the selected order is smaller than the order of the underlying surface, it will result in an underfitting, which smooths out the fine details and affects the accuracy of output normals. Otherwise, overfitting makes the algorithm sensitive to noise and brings instability to the normal estimation [55], as illustrated in Fig. 1(c).

To address these issues, we propose a novel network called *HSurf-Net* for unoriented normal estimation. It makes full use of the powerful learning ability of the neural network to implicitly learn hyper surfaces. The hyper surfaces are represented by MLP layers whose parameters interpret the geometric structures in a high dimensional feature space. The advantage of our hyper surfaces is to adaptively fit more complex point patterns in a robust way. Based on the learned hyper surfaces, we introduce a *Hyper Surface Fitting* module to directly predict normal vectors (see Fig. 1(b)). This module learns from the well extracted point features and gets the surface representation parameters optimized in a data-driven manner, rather than explicitly fitting 3D planes/surfaces by a polynomial function with a predefined order in current methods. Moreover, to avoid the selection of neighborhood scale and construct a noise-less feature space, we design two network modules called *Space Transformation* and *Relative Position Encoding*. They can cover local, small and large scales to enhance the extraction of structure-aware and multi-scale features. Overall, the combination of these modules extracts the discriminative geometric information and avoids the issues caused by explicit polynomial surface fitting, thus improving the performance of the normal estimation framework. We conduct evaluation experiments on the synthetic shape dataset, the real-world indoor and outdoor scene datasets. HSurf-Net significantly outperforms other baselines on the challenging cases in these benchmarks, and also shows a strong generalization capability on real-world LiDAR data. Extensive ablation experiments validate the effectiveness of each component that contributes to the final results.

Our main contributions can be summarized as follows.

- A technique for representing polynomial surfaces as hyper surfaces, which are parameterized by MLP layers.
- A Hyper Surface Fitting in a high dimensional feature space to optimize the surface representation for point cloud normal estimation, which brings more robustness and higher accuracy.

- A Space Transformation module and a Relative Position Encoding module to map 3D point clouds into the feature space. Their combination can fully explore the local geometry and extract features from different neighborhood scales.

## 2 Related Work

**Traditional Normal Estimation**. The classic point cloud normal estimation methods use plane fitting techniques, such as unweighted Principle Component Analysis (PCA) [21] and Singular Value Decomposition (SVD) [45]. They analyze the covariance matrix in a local patch around a point and define its normal as the eigenvector corresponding to the smallest eigenvalue. Subsequently, a variety of PCA-variants [1, 39, 38, 28, 24] have been proposed to improve the accuracy. In order to preserve sharp features and retain more details, a lot of improvements were made to extract normals by using Voronoi cells [3, 37], Voronoi-PCA [14, 2], Hough transform [9] and edge-aware sampling [25]. In addition, some methods utilize more complex surface reconstruction techniques, such as Moving Least Squares (MLS) [31], jet fitting [12], local spherical fitting [16] and multi-scale kernel [4]. The aforementioned approaches come with certain assumptions or specific observations, and hold theoretical guarantees on approximation and robustness. Besides, these approaches need a fine-tuned set of parameters according to the input point clouds.

**Learning-based Normal Estimation**. (1) *Regression based methods*. Initially, some learning-based methods propose to directly regress normal vectors from raw point clouds. HoughCNN [10] transforms 3D points into a 2D grid representation via Hough transform, and then use a CNN to select a normal direction from the accumulator in Hough space. Similarly, Roveri *et al*. [43] define a grid-like regular input to learn point normals. Lu *et al*. [33] project each point into a 2D height map by computing the distances between points and the PCA fitted plane. These methods map the unstructured point cloud data into a regular domain, which may lose or alter the 3D geometric information. Instead, the following approaches learn from raw point clouds. PCPNet [17] applies the PointNet architecture [41] in a local patch-based multi-scale form to estimate normals and curvatures from point clouds. Hashimoto *et al*. [20] propose a two-branch network that extracts local and spatial features, which are integrated to learn normals. Nesti-Net [7] proposes to parameterize the local field as a multi-scale feature vector, then uses a mixture-of-experts architecture [26] to find the optimal neighborhood scale around each point, but it is very time consuming. Based on PCPNet, Zhou *et al*. [53] introduce an extra plane feature constraint and a multi-scale neighborhood selection strategy to improve the performance. Refine-Net [51, 52] first computes an initial normal at each point by searching an ideal fitting patch. Then, a refinement network is used to obtain the final optimal normals by incorporating the learned local point features and the constructed height map features. (2) *Surface fitting based methods*. Recent normal estimation methods try to combine traditional plane/surface fitting techniques with learning-based methods. Lenssen *et al*. [30] propose a fast algorithm that utilizes an adaptive anisotropic kernel to iteratively refine a weighted least squares plane fitting. MTRNet [11] proposes a differentiable RANSAC-like module to predict a latent tangent plane. DeepFit [6] and AdaFit [55] both employ a Taylor expansion to describe the local surface and use the PointNet to predict point-wise weights for a weighted least squares surface fitting. In addition, AdaFit proposes to use a novel layer to aggregate features from multiple neighborhood sizes and add offset to point coordinates to further improve the results. Similarly, Zhou *et al*. [54] use the learned weights and features to select local top-k points and update the point positions. Zhang *et al*. [49] introduce geometric weight guidance to provide more reliable inlier points for plane fitting.

Compared to the learning-based normal regression methods, e.g. PCPNet [17], Nesti-Net [7] and Refine-Net [52], which focus on learning a general mapping from the inputs to the ground-truth normals, our approach improves the normal estimation performance by deploying a hyper surface fitting to explore geometric prior supports in high dimensional space. Compared to the learning-based surface fitting methods, e.g. DeepFit [6] and AdaFit [55], which solve a 3D space based polynomial function with a predefined order to get normals, we effectively improve the results by learning hyper surfaces in a feature space.

## 3 Method

Given a local point set $P = \{p_i | i = 1, ..., N\}$ centralized at a query point $p$, our algorithm aims to estimate the unoriented normal $\mathbf{n}_p$ of point $p$. Fig. 2 shows an overview of the proposed approach.

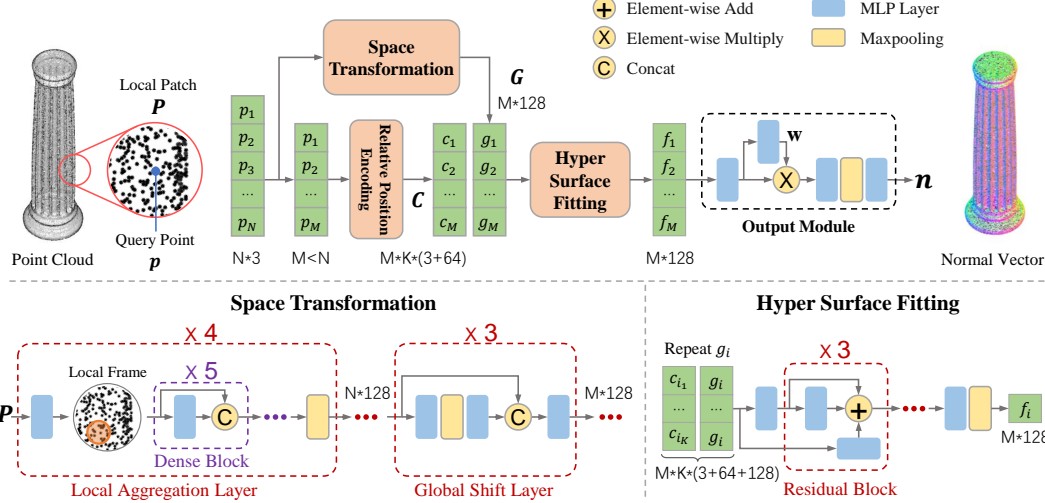

Figure 2: The architecture of HSurf-Net for point cloud normal estimation.

First, to remove unnecessary degrees of freedom from the input data space and lower the learning difficulty, we normalize each point coordinate with its patch radius and rotate the points into a local coordinate system defined by the PCA. Then, the Space Transformation module extracts a *global location code $G$* for each point (Sec. 3.3). Moreover, local frames are formulated at point $p_i$ based on spatial coordinates and are used to compute a *condition code $C$* by a Relative Position Encoding module (Sec. 3.4). After that, we perform the hyper surface fitting in a high dimensional feature space (Sec. 3.2). Finally, we recover the 3D normal vectors from fitting results by an Output Module.

## 3.1 Preliminary

We first briefly review the formulas and mathematical notations of the explicit surface fitting with a polynomial function with a predefined order. A smooth surface can be locally formulated as the graph of a bivariate height function $f(x, y)$ about the $z$-axis that is not in the *tangent space* [44, 29]. An $n$-order Taylor expansion of the height function over a surface is given by

$$f(x,y) = J_{\beta,n}(x,y) + O(||(x,y)||^{n+1}), \text{ where } J_{\beta,n}(x,y) = \sum_{k=0}^{n}\sum_{j=0}^{k} \beta_{k-j,j}x^{k-j}y^{j}. \quad (1)$$

The polynomial function $J_{\beta,n} : \mathbb{R}^2 \to \mathbb{R}$ is the truncated Taylor expansion called $n$-jet [12]. $\beta$ is the $n$-jet coefficient vector and involves $N_n = (n+1)(n+2)/2$ terms. $O(\cdot)$ denotes the remainder in Taylor's multivariate formula. The predefined order $n$ determines the complexity of the fitted surface.

Consider a collection of points $P = \{p_i = (x_i, y_i, z_i) | i = 1, ..., N\}$ around its origin $p$ on a sampled smooth surface $S$, and the height function $z = f(x, y)$ given by Eq. (1) in the defined coordinate system, we can investigate the $n$-order differential property of the surface $S = f(x_i, y_i)$ at point $p_i$ by interpolating $S$ using a bivariate $n$-jet $J_{\alpha,n}(x, y)$, such that

$$f(x_i, y_i) \simeq J_{\alpha,n}(x_i, y_i), \ \ \forall i = 1, ..., N, \quad (2)$$

where $\alpha$ means the coefficient of the jet sought. Thus, the coefficient $\alpha$ is the solution of the $n$-jet surface fitting, denoted as $N_n$-vector $\alpha = (\alpha_{0,0}, \alpha_{1,0}, \alpha_{0,1}, ..., \alpha_{0,n})^T$. To find the solution, the least squares approximation strategy is used to minimize the sum of the square errors between the jet value and the height function over the point set $P$

$$J_{\alpha,n}^* = \underset{\alpha}{\operatorname{argmin}} \sum_{i=1}^{N}(f(x_i, y_i) - J_{\alpha,n}(x_i, y_i))^2. \quad (3)$$

Then the surface fitting problem of Eq. (2) is described as finding the least squares solution of a homogeneous system of linear equations. Once the $n$-jet coefficient $\alpha$ is solved, the normal vector at point $p$ on the fitted surface is computed by

$$\mathbf{n}_p = h(\alpha) = (-\alpha_{1,0}, -\alpha_{0,1}, 1)/\sqrt{1 + \alpha_{1,0}^2 + \alpha_{0,1}^2} . \quad (4)$$

## 3.2 Hyper Surface Fitting

The explicit surface fitting method in Sec. 3.1 requires a predefined polynomial function with an order $n$ and its performance is susceptible to noise and outliers, as shown in Sec. 4.3. Motivated by the rapidly developed data-driven approaches, which excel at adaptively learning a fitting model that describes the pattern of the provided noisy data [13, 32, 36, 48], we propose to implicitly learn hyper surfaces in the *feature space* using a *Hyper Surface Fitting* technique. We expand the 3D point coordinates $(x, y, z)$ into high dimensional features $(G, C, F)$, and $F \sim \mathcal{F}(G, C), F \in \mathbb{R}^c$. Then the new feature-based formulation for the polynomial function $J_{\alpha,n}(x, y)$ in Eq. (2) is given by (see supplementary materials for the derivation)

$$\mathcal{N}_{\theta,\tau}(G, C) = \sum_{k=0}^{\tau} \sum_{j=0}^{k} \theta_{k-j,j} \, \mathbf{g}_{k-j} \mathbf{c}_j = \theta \, [G : C], \tag{5}$$

where [ : ] means feature fusion operation, such as *concatenation*. $G \in \mathbb{R}^c$ and $C \in \mathbb{R}^c$ are high dimensional features of the 3D point clouds extracted by two different modules, which are introduced in the following two sections. Here, both the parameter $\tau$ and the coefficient $\theta$ are parameters of an MLP-based module, which is designed as a sequence of skip-connected Residual Block units (see Fig. 2). Similar with Eq. (3), the bivariate function $\mathcal{N}_{\theta,\tau}(G, C)$ aims to map each feature pair $(G_i, C_i)$ to their true value $\mathcal{F}(G_i, C_i) \in \mathbb{R}^c$ in the feature space

$$\mathcal{N}_{\theta,\tau}^* = \underset{\theta,\tau}{\operatorname{argmin}} \sum_{i=1}^{N} \|\mathcal{N}_{\theta,\tau}(G_i, C_i) - \mathcal{F}(G_i, C_i)\|^2. \tag{6}$$

In contrast to compute point normals formulaically by the $n$-jet coefficients as in Eq. (4), we recover the normal vectors $\mathbf{n}$ from the $c$-dimensional hyper surface using a function $\mathcal{H} : \mathbb{R}^c \to \mathbb{R}^3$, which is applied to the fitting surface and the surface of fitting sought, i.e. the ground-truth surface

$$\mathcal{N}_{\theta,\tau}^* = \underset{\theta,\tau}{\operatorname{argmin}} \sum_{i=1}^{N} \|\mathcal{H}(\mathcal{N}_{\theta,\tau}(G_i, C_i)) - \mathcal{H}(\mathcal{F}(G_i, C_i))\|^2. \tag{7}$$

Then we get the optimization function about the normals

$$\mathcal{N}_{\theta,\tau}^* = \underset{\theta,\tau}{\operatorname{argmin}} \sum_{i=1}^{N} \|\mathbf{n}_i - \hat{\mathbf{n}}_i\|^2, \tag{8}$$

where $\mathbf{n}$ and $\hat{\mathbf{n}}$ are the output unoriented normal and the ground-truth, respectively. Finally, the normal of the query point $p$ is formulated as the weighted maxpooling of its neighborhoods (see *Output Module* in Fig. 2)

$$\mathbf{n}_p = \dot{\mathbf{n}}_p / \|\dot{\mathbf{n}}_p\|, \ \dot{\mathbf{n}}_p = \mathcal{H}(\operatorname{MAX}\{w_i \, \mathcal{N}_{\theta,\tau}(G_i, C_i) | i = 1, ..., N\}), \tag{9}$$

where $\operatorname{MAX}\{\cdot\}$ means maxpooling, $w_i = \operatorname{sigmoid}(\Psi(\mathcal{N}_{\theta,\tau}(G_i, C_i)))$ is the weight, $\mathcal{H}$ and $\Psi$ are MLPs. We experimentally find that $sin$ distance $\|\hat{\mathbf{n}}_p \times \mathbf{n}_p\|$ is more suitable for measuring the vector difference and guiding the estimated normal to match the ground-truth (see Sec. 4.3). Our method adaptively learns the optimal hyper surface $\mathcal{N}_{\theta,\tau}^*$ from high dimensional features, and it is more robust than the $n$-jet fitting, which fits a single type of surface from 3D points with a constant order.

## 3.3 Space Transformation

The previous methods usually use the PointNet-like architectures to learn features from point clouds. Since PointNet is inadequate for encoding local structures, we design a feature extraction module called *Space Transformation*, which learns from local neighborhood, small and large patch scales. This module provides a noise-less *global location code G* for the hyper surface fitting, and it consists of a sequence of Local Aggregation Layer units and Global Shift Layer units (see Fig. 2). We repeatedly employ each kind of unit on different levels of feature representation details. The raw point clouds are often noisy and the points have position offsets relative to the noise-free points, which will lead to deviation in the learned features. The Local Aggregation Layer delivers a smooth filtered relative feature of each point by the cascaded local frame construction and maxpooling operation. Since the relative features only describe the local structures, we design a Global Shift Layer to provide the final features with global position information in the feature space by fusing global features. Specifically, we explore the global information from different scales, as shown in Fig. 3.

In the *Local Aggregation Layer*, we first convert the input points to a fixed dimension. Then, we group the local neighborhood features at each point by the 3D spatial distance based kNN search, and refine each grouped feature via a chain of Dense Block units [23]. Finally, we compute an order-invariant per-point feature via maxpooling [42]. In the Dense Block, we use the skip-connection to leverage features extracted across different layers. In this way, the information of different layers is combined via intra-level connections inside the unit, which realizes the information reuse, thereby improving the learning ability of the network.

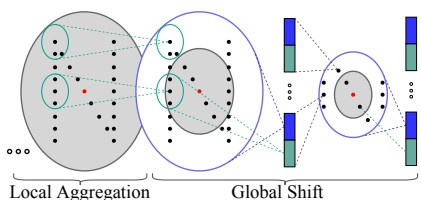

Figure 3: Space Transformation module.

In the *Global Shift Layer*, we provide the localization of points in the entire patch feature space by fusing global features extracted from multiple neighborhood scales of the query point $p$. Extracting multi-scale features has become an effective strategy to further boost the normal estimation performance [10, 7, 17, 53, 11, 55]. The small scale results in a more accurate description of the details, while the large scale provides more information about the underlying geometries. In summary, our scale-aware Global Shift Layer is formulated as (see the blue part in Fig. 3)

$$G_{s+1,i} = \mathcal{U}_s(\mathcal{V}_s(\text{MAX}\{G_{s,j}|j=1,..,N_s\}),\ G_{s,i}),\ i=1,...,N_{s+1}\ , \tag{10}$$

where $\mathcal{U}, \mathcal{V}$ are MLP layers, $G_{s,i}$ is the per-point feature at scale $s$. $\text{MAX}\{\cdot\}$ means performing feature maxpooling over all neighboring points of $p$ with size $N_s$ and $N_{s+1} \leqslant N_s$.

### 3.4 Relative Position Encoding

Position encoding is important for the transformer architecture [47, 18, 50], allowing the self-attention mechanism to capture sequence order information of input tokens. We borrow the idea and use it here to extract a *condition code* $C$ from the local geometry of point cloud data. Contrast to the traditional position encoding, which manually craft for language sequence or image grid based on sine and cosine functions, we design a parameterized and learnable encoding scheme which is trained together with the whole model. Given a point $p_i \in \{p_i|i=1, ..., M\}$ and its neighborhoods $\{p_i{}^j|j=1, ..., K\}$, our learning-based position encoding function $\phi$ using relative coordinates is formulated as

$$C_i{}^j = \phi(p_i{}^j - p_i,\ \mathcal{E}(p_i{}^j - p_i)), \tag{11}$$

where $p_i{}^j - p_i$ denotes the relative position in a local frame, and $M = N/4$ neighboring points of $p$ are used for encoding. The encoding functions $\mathcal{E}$ and $\phi$ are MLP layers. Experimental results show that our encoding scheme outperforms the traditional position encoding in the normal estimation task.

### 3.5 Loss Functions

To predict normal vectors that match the ground-truth as close as possible, we minimizes the $sin$ loss between the output unoriented normal $\mathbf{n}_p$ and the ground-truth normal $\hat{\mathbf{n}}_p$ at the query point $p$

$$L_1 = \|\hat{\mathbf{n}}_p \times \mathbf{n}_p\|. \tag{12}$$

Meanwhile, we also adopt a weight loss term similar to [49]

$$L_2 = \frac{1}{N} \sum_{i=1}^{N} (w_i - \hat{w}_i)^2, \tag{13}$$

where $w$ is the generated point weights in the output module. $\hat{w}_i = \exp(-(p_i \cdot \hat{\mathbf{n}}_p)^2/\delta^2)$ and $\delta = \max(0.05^2,\ 0.3 \sum_{i=1}^{N} (p_i \cdot \hat{\mathbf{n}}_p)^2/N)$, where $p_i, \hat{\mathbf{n}}_p \in \mathbb{R}^3$. In total, the final loss is given by

$$L = \alpha_1 L_1 + \alpha_2 L_2, \tag{14}$$

where $\alpha_1 = 0.1$ and $\alpha_2 = 1.0$ are weighting factors.

## 4 Experiments

**Datasets and Settings**. We first compare our method with other baselines using the same train/test data of PCPNet shape dataset [17]. To evaluate the generalization ability of each method on the

Table 1: Normal angle RMSE results on the PCPNet and SceneNN dataset. Sorted by the average values on the PCPNet dataset. Lower is better. ∗ means the code is not available or uncompleted.

| Category | PCPNet Dataset | | | | | | | SceneNN Dataset | | |
| | None | Noise $\sigma$ 0.12% | 0.6% | 1.2% | Density Stripes | Gradient | **Average** | Clean | Noise | **Average** |
|---|---|---|---|---|---|---|---|---|---|---|
| Jet [12] | 12.35 | 12.84 | 18.33 | 27.68 | 13.39 | 13.13 | 16.29 | 15.17 | 15.59 | 15.38 |
| PCA [21] | 12.29 | 12.87 | 18.38 | 27.52 | 13.66 | 12.81 | 16.25 | 15.93 | 16.32 | 16.12 |
| PCPNet [17] | 9.64 | 11.51 | 18.27 | 22.84 | 11.73 | 13.46 | 14.58 | 20.86 | 21.40 | 21.13 |
| Zhou et al.∗ [53] | 8.67 | 10.49 | 17.62 | 24.14 | 10.29 | 10.66 | 13.62 | - | - | - |
| Nesti-Net [7] | 7.06 | 10.24 | 17.77 | 22.31 | 8.64 | 8.95 | 12.49 | 13.01 | 15.19 | 14.10 |
| Lenssen et al. [30] | 6.72 | 9.95 | 17.18 | 21.96 | 7.73 | 7.51 | 11.84 | 10.24 | 13.00 | 11.62 |
| DeepFit [6] | 6.51 | 9.21 | 16.73 | 23.12 | 7.92 | 7.31 | 11.80 | 10.33 | 13.07 | 11.70 |
| MTRNet∗ [11] | 6.43 | 9.69 | 17.08 | 22.23 | 8.39 | 6.89 | 11.78 | - | - | - |
| Refine-Net [52] | 5.92 | 9.04 | 16.52 | 22.19 | 7.70 | 7.20 | 11.43 | 18.09 | 19.73 | 18.91 |
| Zhang et al.∗ [49] | 5.65 | 9.19 | 16.78 | 22.93 | 6.68 | 6.29 | 11.25 | 9.31 | 13.11 | 11.21 |
| Zhou et al.∗ [54] | 5.90 | 9.10 | 16.50 | 22.08 | 6.79 | 6.40 | 11.13 | - | - | - |
| AdaFit [55] | 5.19 | 9.05 | 16.45 | 21.94 | 6.01 | 5.90 | 10.76 | 8.39 | 12.85 | 10.62 |
| Ours | **4.17** | **8.78** | **16.25** | **21.61** | **4.98** | **4.86** | **10.11** | **7.55** | **12.23** | **9.89** |

(a) PCPNet Dataset        (b) SceneNN Dataset

Figure 4: Normal error AUC results on the PCPNet and SceneNN dataset. X-axis is the angle threshold in degree and Y-axis is the percentage of good point (PGP) normals under a given threshold.

real-world scene data, we employ the models pretrained on the PCPNet dataset to report results on the indoor SceneNN dataset [22] and the outdoor Semantic3D dataset [19]. For PCPNet dataset, we follow the experimental setup in [17] including the train/test set split, noise level, density variations and the data augmentation. We randomly sample 1000 query points for each shape during training. We set the initial point cloud patch size $N = 700$ and scale set $N_s = \{N, N/2, N/4\}$. The number of kNN points in the Space Transformation is 16 and in the encoding module $K = 16$. We use Adam optimizer with initial learning rate $5 \times 10^{-4}$ and the learning rate is decayed to $\times 0.2$ of the latest value after every 200 epochs. The model is trained with a batch size of 100 in 900 epochs on an NVIDIA 2080 Ti GPU.

**Evaluation Metrics**. We adopt the widely-used angular Root Mean Squared Error (RMSE) between the predicted normal and the ground-truth to evaluate the normal estimation results. We also use the Area Under the Curve (AUC) to analyze the error distribution of the predicted normal. It is presented by the Percentage of Good Points (PGP) metric, which measures the percentage of point normals with errors below different angle thresholds.

## 4.1 Results on Synthetic Shape Dataset

Table 1 reports numerical comparison in terms of RMSE on the PCPNet dataset [17]. Fig. 4(a) shows the normal error AUC results. It can be seen that our method outperforms both the traditional

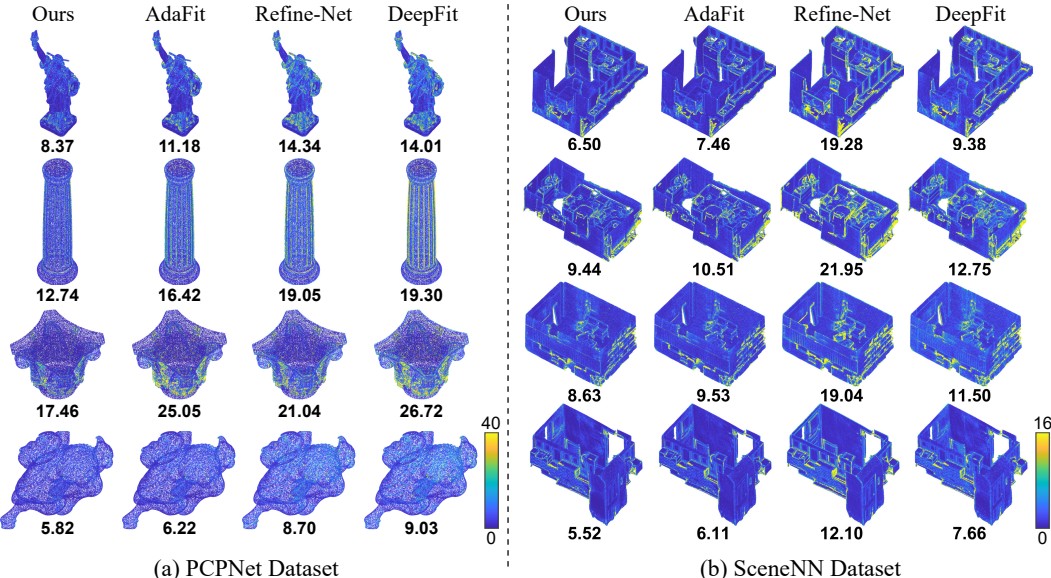

(a) PCPNet Dataset        (b) SceneNN Dataset

Figure 5: Error visualization results on the PCPNet complex shapes and SceneNN dataset. The number under each point cloud is its RMSE. The point color is the angular error mapped to a heatmap.

methods and the learning-based methods across all noise levels and density variations. The results also show that the performance of all methods degrades substantially as the noise level increases, but the influence of density variations is much smaller than the noise. Fig. 5(a) shows the visual comparison on different shapes, where the point clouds are rendered with a normal angle error map. We can see that errors mostly occur in regions with complex geometry or high curvature, e.g. edges and corners. Our method shows smallest errors on these regions due to its hyper surface fitting in a high dimensional feature space, enabling it to adapt to the different local geometry and disregard irrelevant outliers.

## 4.2 Results on Indoor and Outdoor Scene Datasets

Table 1 presents numerical comparison with the state-of-the-art results in terms of RMSE on the SceneNN dataset [22]. Fig. 4(b) shows the normal angle error AUC results. Fig. 5(b) visualizes the qualitative comparison of errors in different room scenes. The quantitative and qualitative comparison results show that our method outperforms both the traditional and the learning-based methods in all scenes and data categories.

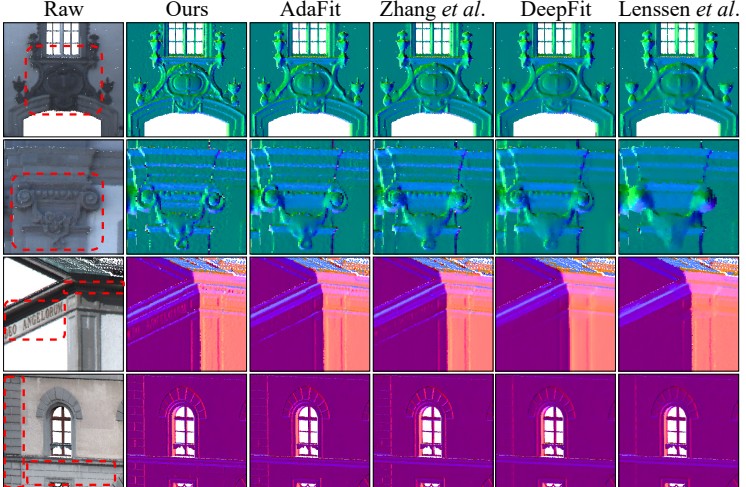

Figure 6: Visual comparison on the Semantic3D dataset. The point normal vectors are mapped to RGB colors.

In Fig. 6, we show visual comparison on the Semantic3D dataset [19] as the absence of the ground-truth normal. We can observe that our method preserves fine details in the complex structures of the building, such as sharp edges, groove between the bricks, pattern and relief on the wall, while other methods behave over-smooth in these areas or give wrong normals. The comparison results demonstrate that our method generalizes well to the real-world LiDAR data.

Table 2: Normal angle RMSE results of ablation studies on the PCPNet dataset. The ablation experiments include: our new designed modules, i.e. (a) Hyper Surface Fitting, (b) Space Transformation, (c) Relative Position Encoding and (d) Output Module, and a hyperparameter, i.e. (e) the input point cloud patch size. Please see the text for more details.

| Ablation | | Hyper Surface | Space Trans. | Position Encoding | Output Module | Noise $\sigma$ | | | | Density | | Average |
|---|---|---|---|---|---|---|---|---|---|---|---|---|
| | | | | | | None | 0.12% | 0.6% | 1.2% | Stripes | Gradient | |
| (a) | Jet $n=1$ | | ✓ | ✓ | ✓ | 7.55 | 9.95 | 16.54 | 21.99 | 8.89 | 7.83 | 12.12 |
| | Jet $n=2$ | | ✓ | ✓ | ✓ | 8.52 | 10.13 | 16.59 | 22.12 | 10.09 | 9.05 | 12.75 |
| | Jet $n=3$ | | ✓ | ✓ | ✓ | 7.50 | 9.70 | 16.68 | 22.22 | 9.01 | 8.07 | 12.19 |
| | Simple | | ✓ | ✓ | ✓ | 4.44 | 8.84 | 16.29 | 21.76 | 5.27 | 5.18 | 10.30 |
| (b) | PointNet-based | ✓ | | ✓ | ✓ | 4.84 | 9.09 | 16.47 | 21.98 | 5.67 | 5.63 | 10.61 |
| | $N'=256$ | ✓ | | ✓ | ✓ | 5.56 | 9.19 | 16.76 | 23.20 | 6.71 | 6.21 | 11.27 |
| | $N'=500$ | ✓ | | ✓ | ✓ | 5.51 | 9.24 | 16.65 | 22.10 | 7.11 | 6.23 | 11.14 |
| | $N'=700$ | ✓ | | ✓ | ✓ | 5.80 | 9.32 | 16.61 | 21.86 | 7.15 | 6.28 | 11.17 |
| (c) | w/ Traditional | ✓ | ✓ | | ✓ | 4.43 | 8.89 | 16.19 | 21.64 | 5.44 | 5.14 | 10.29 |
| | w/o Encoding | ✓ | ✓ | | ✓ | 4.66 | 8.80 | 16.32 | 21.77 | 5.60 | 5.37 | 10.42 |
| (d) | w/o Weight | ✓ | ✓ | ✓ | | 5.04 | 8.98 | 16.48 | 21.80 | 6.14 | 5.85 | 10.71 |
| | w/ Loss $L_{con}$ | ✓ | ✓ | ✓ | ✓ | 4.35 | 8.90 | 16.20 | 21.77 | 5.34 | 5.17 | 10.29 |
| | w/ Loss $L_{MSE}$ | ✓ | ✓ | ✓ | ✓ | 10.96 | 12.68 | 19.76 | 24.76 | 13.26 | 11.29 | 15.45 |
| (e) | Full $N=600$ | ✓ | ✓ | ✓ | ✓ | 4.25 | **8.69** | 16.20 | 21.78 | 5.11 | 4.92 | 10.16 |
| | Full $N=800$ | ✓ | ✓ | ✓ | ✓ | 4.30 | 8.73 | **16.19** | **21.53** | 5.08 | 5.04 | 10.15 |
| | **Final** $N=700$ | ✓ | ✓ | ✓ | ✓ | **4.17** | 8.78 | 16.25 | 21.61 | **4.98** | **4.86** | **10.11** |

## 4.3 Ablation Studies

**(a) Hyper Surface Fitting**. To validate the effectiveness of the proposed hyper surface fitting technique, we compare the performance of different models with the $n$-jet fitting and our hyper surface fitting. The same backbone network (i.e. space transformation and position encoding) is used in this experiment, but the inputs of the jet fitting based models are 3D points and point-wise weights, which are predicted by the backbone network. We also evaluate a model that replaces the residual block in our hyper surface fitting module with a concatenation operation, and we call it "Simple". The results are shown in Table 2(a), we can see that the performance of the models with jet fitting is sensitive to the polynomial order $n$, while the hyper surface fitting can effectively improve the performance. Moreover, a simple 1-order may be enough for the $n$-jet fitting in the normal estimation task.

**(b) Space Transformation**. Our Space Transformation is composed of Local Aggregation Layer units and Global Shift Layer units. In Table 2(b), we conduct feature extraction experiments using two types of the Space Transformation: 1) with a PointNet based aggregation layer; 2) without the multi-scale Global Shift Layer and with a fixed scale $N'$. The results show that our novel components can effectively boost the normal estimation performance.

**(c) Encoding Module**. In Table 2(c), we provide the results with the traditional position encoding in [47]. We also provide the results of a model that directly uses 3D point coordinates as the condition code $C$ without any encoding processing. The results show that our learnable encoding scheme can improve the performance, and also remove the requirement of selecting specific encoding functions in the traditional one.

**(d) Output Module**. We use a weighted maxpooling operation in Eq. (9) to recover the query point normal $\mathbf{n}_p$ from the hyper surface fitting results. For ablation, we estimate the normal without using the weight, i.e., $\mathbf{n}_p = \mathcal{H}(\text{MAX}\{\mathcal{N}_{\theta,\tau}(G_i, C_i)|i = 1, ..., N\})$. Moreover, we also evaluate other two models: 1) trained with an additional weighted neighborhood consistency loss $L_{con} = \frac{1}{N}\sum_{i=1}^{N} w_i|\hat{\mathbf{n}}_i \times \mathbf{n}_i|$ where the neighboring point normals are predicted by another function $\mathbf{n}_i = \mathcal{H}'(\mathcal{N}_{\theta,\tau}(G_i, C_i))$; 2) trained with the MSE loss $L_{MSE}$ instead of the $sin$ loss. The results are shown in Table 2(d), and we can observe that: 1) the weight can improve the results; 2) the further added consistency loss $L_{con}$ is not helpful for the final results, which means that we only need the query point normal in each point cloud patch and omit the neighboring point normals (also reduce network parameters); 3) the loss $L_{MSE}$ can not guide the model to learn accurate normals.

**(e) Input Patch Size**. In the previous experiments, we set the input point cloud patch size to 700. In Table 2(e), we evaluate two full models with different patch sizes $N=600$ and $N=800$. The larger size brings better results than the smaller size under large noise, but takes more time and memory.

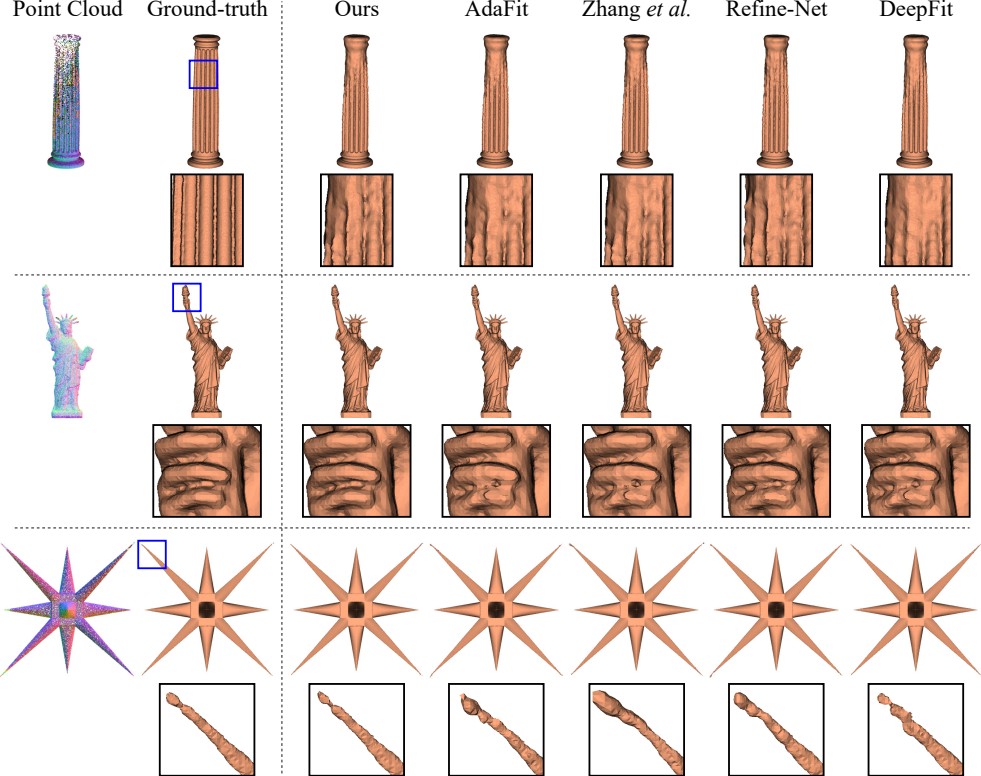

Figure 7: Visual comparison of the reconstructed surface using the normals estimated by different methods. A local enlarged view is provided under each shape.

## 4.4 Application to Surface Reconstruction

We adopt the Poisson surface reconstruction [27] to reconstruct the object surface from the point cloud with point-wise normals estimated by different methods. Fig. 7 shows the reconstructed surfaces of different objects. We can see that our estimated normals can facilitate the reconstruction algorithm to produce more accurate and complete surfaces than other baseline methods. In the supplementary materials, we provide more evaluation results and subsequent applications of normal estimation.

## 5 Conclusion

In this paper, we solved several problems in existing point cloud normal estimation methods. We propose to directly regress point normal vectors through a novel hyper surface fitting technique, which implicitly learns hyper surfaces from noisy 3D points in a high dimensional feature space, rather than fitting a geometric plane or surface using a polynomial function with a predefined order. Furthermore, we propose two novel modules, i.e. Space Transformation and Relative Position Encoding, to transform 3D point clouds into the feature space from different spatial dimensions. We show that these designs are more effective than their respective counterparts. As a result, our HSurf-Net reduces the influence of noise and outliers on the fitting process and improves the robustness and accuracy of normal estimation. Extensive experiments prove the effectiveness of HSurf-Net, and demonstrate that it produces visually and quantitatively better results than the state-of-the-art methods. The future works include exploring more effective point cloud representations, and more efficient local feature extraction for normal estimation.

## 6 Acknowledgement

This work was supported by National Key R&D Program of China (2022YFC3800600, 2020YFF0304100), the National Natural Science Foundation of China (62272263, 62072268), and in part by Tsinghua-Kuaishou Institute of Future Media Data.

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
