# HSurf-Net: Normal Estimation for 3D Point Clouds by Learning Hyper Surfaces
## — Supplementary Material —

**Qing Li**[1]    **Yu-Shen Liu**[1][*]   **Jin-San Cheng**[2]    **Cheng Wang**[3]
**Yi Fang**[4]    **Zhizhong Han**[5]
[1]School of Software, BNRist, Tsinghua University, Beijing, China
[2]Academy of Mathematics and System Sciences, Chinese Academy of Sciences, Beijing, China
[3]School of Informatics, Xiamen University, Xiamen, China
[4]Center for Artificial Intelligence and Robotics, New York University Abu Dhabi, Abu Dhabi, UAE
[5]Department of Computer Science, Wayne State University, Detroit, USA
{leoqli, liuyushen}@tsinghua.edu.cn   jcheng@amss.ac.cn
cwang@xmu.edu.cn   yfang@nyu.edu   h312h@wayne.edu

## 1   Notations

| | |
|---|---|
| $n$ | The order of polynomial function |
| $\mathbf{n}$ | The point normal vector |
| $P = \{p_i \mid i = 1, ..., N\}$ | Input point cloud patch |
| $p = (x, y, z)$ | A point and its coordinate |
| $f(x, y)$ | The height function in 3D space |
| $S$ | A smooth surface in 3D space |
| $O(\cdot)$ | The remainder in Taylor's multivariate formula |
| $\alpha, \beta$ | The coefficient of polynomial function |
| $J, J^*$ | The polynomial function and its optimal solution |
| $\mathcal{N}, \mathcal{N}^*$ | The hyper surface and its optimal solution |
| $\tau$ | The order of hyper surface (i.e., parameters of an MLP) |
| $\theta$ | The coefficient of hyper surface (i.e., parameters of an MLP) |
| $(G, C, F)$ | The high dimensional feature space |
| $G$ | The global location code from Space Transformation module |
| $C$ | The condition code from Relative Position Encoding module |
| $\mathcal{F}$ | The height function in feature space |
| $N, M$ | The number of points in the point cloud patch |
| $N_n$ | The number of terms in the coefficient |
| $N_s$ | The neighborhood size of the query point $p$ at scale $s$ |
| $N_k$ | The per-point neighborhood size in the Relative Position Encoding |
| $\mathcal{H}, \Psi$ | MLP Layers in the Hyper Surface Fitting |
| $\mathcal{U}, \mathcal{V}$ | MLP Layers in the Global Shift Layer |
| $\mathcal{E}, \phi$ | MLP Layers in the Relative Position Encoding |
| $\mathrm{MAX}\{\cdot\}$ | Maxpooling |

---

[*]Correspondig author

36th Conference on Neural Information Processing Systems (NeurIPS 2022).

## 2 Additional Details

### 2.1 Derivation

For Eq.(5) in the paper, we provide a more detailed derivation process to clarify its relationship to the designed networks. In order to expand the polynomial surface fitting in 3D dimensional space into the high dimensional feature space using a neural network with parameter $\Theta$, we define $f_1(g^\omega) := \mathbf{g}$ and $f_2(c^\upsilon) := \mathbf{c}$, where $f$ means MLP layer. Then, the multiplication of real numbers $g^\omega \odot c^\upsilon$ in the polynomial function is represented as $\mathbf{g} \boxdot \mathbf{c}$, i.e., $g^\omega \odot c^\upsilon := \mathbf{g} \boxdot \mathbf{c}$, and the orders $\omega, \upsilon \in [0, 1, ..., \tau]$. For the symbol $\boxdot$, we experimentally choose the commonly used feature concatenation operation, hence

$$
\begin{aligned}
\mathcal{N}_{\theta,\tau}(G, C) &= \sum_{k=0}^{\tau} \sum_{j=0}^{k} \theta_{k-j,j} \; g^{k-j} c^j \\
&= \theta_{0,0} + \theta_{1,0}g + \theta_{0,1}c + \cdots + \theta_{1,\tau-1}gc^{\tau-1} + \theta_{0,\tau}c^\tau \\
&= [\theta_{0,0}, \theta_{1,0}, \theta_{0,1}, \cdots, \theta_{1,\tau-1}, \theta_{0,\tau}] \begin{bmatrix} 1 \\ g \\ c \\ \vdots \\ gc^{\tau-1} \\ c^\tau \end{bmatrix} \\
&= [\theta_{0,0}, \theta_{1,0}, \theta_{0,1}, \cdots, \theta_{1,\tau-1}, \theta_{0,\tau}] \left( \begin{bmatrix} g^0 \\ g^1 \\ g^0 \\ \vdots \\ g^1 \\ g^0 \end{bmatrix} \odot \begin{bmatrix} c^0 \\ c^0 \\ c^1 \\ \vdots \\ c^{\tau-1} \\ c^\tau \end{bmatrix} \right) \quad \text{with } f \\
&:= \Theta \left( \begin{bmatrix} \mathbf{g}_1 \\ \mathbf{g}_2 \\ \mathbf{g}_3 \\ \vdots \\ \mathbf{g}_{N_\tau-1} \\ \mathbf{g}_{N_\tau} \end{bmatrix} \boxdot \begin{bmatrix} \mathbf{c}_1 \\ \mathbf{c}_2 \\ \mathbf{c}_3 \\ \vdots \\ \mathbf{c}_{N_\tau-1} \\ \mathbf{c}_{N_\tau} \end{bmatrix} \right) \\
&= \Theta(G \boxdot C).
\end{aligned}
$$

Then, the final bivariate function used in our hyper surface fitting is $\mathcal{N}_{\theta,\tau}(G, C) = \Theta(G \boxdot C)$, where $G$ and $C$ are high dimensional features of the 3D point clouds extracted by the two different modules, which are introduced in Sec.3.3 and Sec.3.4 of the paper, respectively.

### 2.2 Max-pooling in Eq. (9) of the Main Paper

In the traditional polynomial surface fitting, we have [5]

$$
\begin{aligned}
J_{\alpha,n}(x, y) = \alpha_{0,0} + (\alpha_{1,0}x + \alpha_{0,1}y) + \frac{1}{2}(\alpha_{2,0}x^2 + 2\alpha_{1,1}xy + \alpha_{0,2}y^2) \\
+ \frac{1}{6}(\alpha_{3,0}x^3 + 3\alpha_{2,1}x^2y + \cdots) + \cdots.
\end{aligned}
\tag{1}
$$

The origin, that is the point of the fitted surface where the estimation is performed, is $(0, 0, \alpha_{0,0})$. The coefficients of the principal terms $(\alpha_{1,0}x + \alpha_{0,1}y)$ in the above polynomial equation are used to calculate the normal of the fitted polynomial surface at the origin

$$
\mathbf{n}_p = h(\alpha) = (-\alpha_{1,0}, -\alpha_{0,1}, 1)/\sqrt{1 + \alpha_{1,0}^2 + \alpha_{0,1}^2}.
$$

The other terms except the principal terms in the polynomial equation are not used in the estimation of the normal. Based on this, we use the max-pooling over all features from the hyper surface fitting

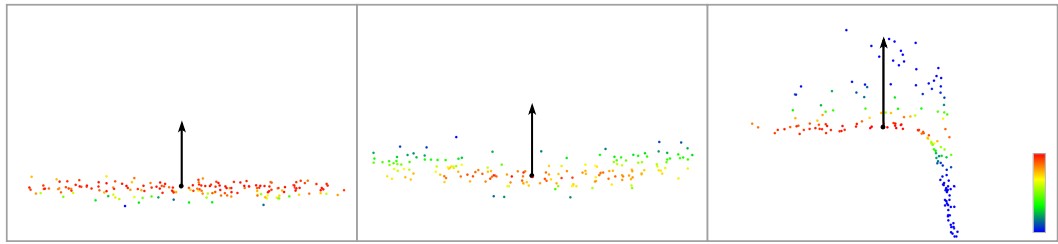

Figure 1: Visualization of the contribution of each 3D point to estimate the normal of the query point (black). The red points are weighted higher than the blue points.

to choose the most prominent feature. Then, we use a liner layer $\mathcal{H}$ to predict the normal of the query point $p$ (i.e. origin)

$$\mathbf{n}_p = \|\mathcal{H}(\text{MAX}\{w_i \, \mathcal{N}_{\theta,\tau}(G_i, C_i)|i = 1, ..., N\})\|,$$

where $\mathcal{N}_{\theta,\tau}(G_i, C_i)$ means the per-point feature result obtained from the hyper surface fitting process.

## 2.3 Datasets

**Synthetic Shape Dataset**. The PCPNet dataset [6] consists of 30 shapes, including high quality scans of figurines, CAD objects, and analytic shapes, with 8 shapes for training and 22 for testing. All shapes are uniformly sampled with 100k points from triangle meshes. Each point cloud is augmented by introducing Gaussian noise with standard deviations of 0.12%, 0.6% and 1.2% of the diagonal length of the bounding box. In addition, two categories of point cloud with different sampling densities (stripe and gradient) are generated for each figurine and CAD object.

**Indoor Scene Dataset**. The SceneNN dataset [9] contains variety indoor scenes collected by a depth camera, and we select four completely scanned scenes as our test data. We compute the ground-truth normal from the provided reconstructed meshes and obtain one million points for each scene by sampling on the mesh. In addition, we generate another point cloud for each scene by adding Gaussian noise with standard deviations of 0.003 to the sampled points.

**Outdoor Scene Dataset**. The Semantic3D dataset [7] provides point clouds captured by a 3D laser scanner in different outdoor environment, and we select a large-scale building scene with about 9.7 million points as our test data. It includes missing data and noise patterns that are very different from the PCPNet and SceneNN dataset. Note that there is no reconstructed mesh data or ground-truth normal for the Semantic3D dataset. For the SceneNN and Semantic3D dataset, in order to decrease useless points for the evaluation of normal estimation and reduce the computation time, we manually remove some isolated points and feature-less flat plane, such as the ground of square. The preprocessed data is used as the test data for all methods and the data will be made publicly available along with our code.

## 2.4 Baseline Method

Our evaluation of point cloud normal estimation covers three types of baselines: (1) the traditional normal estimation methods, e.g. PCA [8] and Jet [4]; (2) the learning-based surface fitting methods, e.g. DeepFit [1] and AdaFit [16]; (3) the learning-based normal regression methods, e.g. PCPNet [6], Nesti-Net [2] and Refine-Net [12].

Since the method of Zhang *et al*. [11] needs the precomputed feature as its input which is not publicly available, we modify the source code to only feed 3D point clouds into their model. The numerical results of Zhou *et al*. [13], MTRNet [3] and Zhou *et al*. [14] on the PCPNet dataset are taken from their papers due to the absence of their source code.

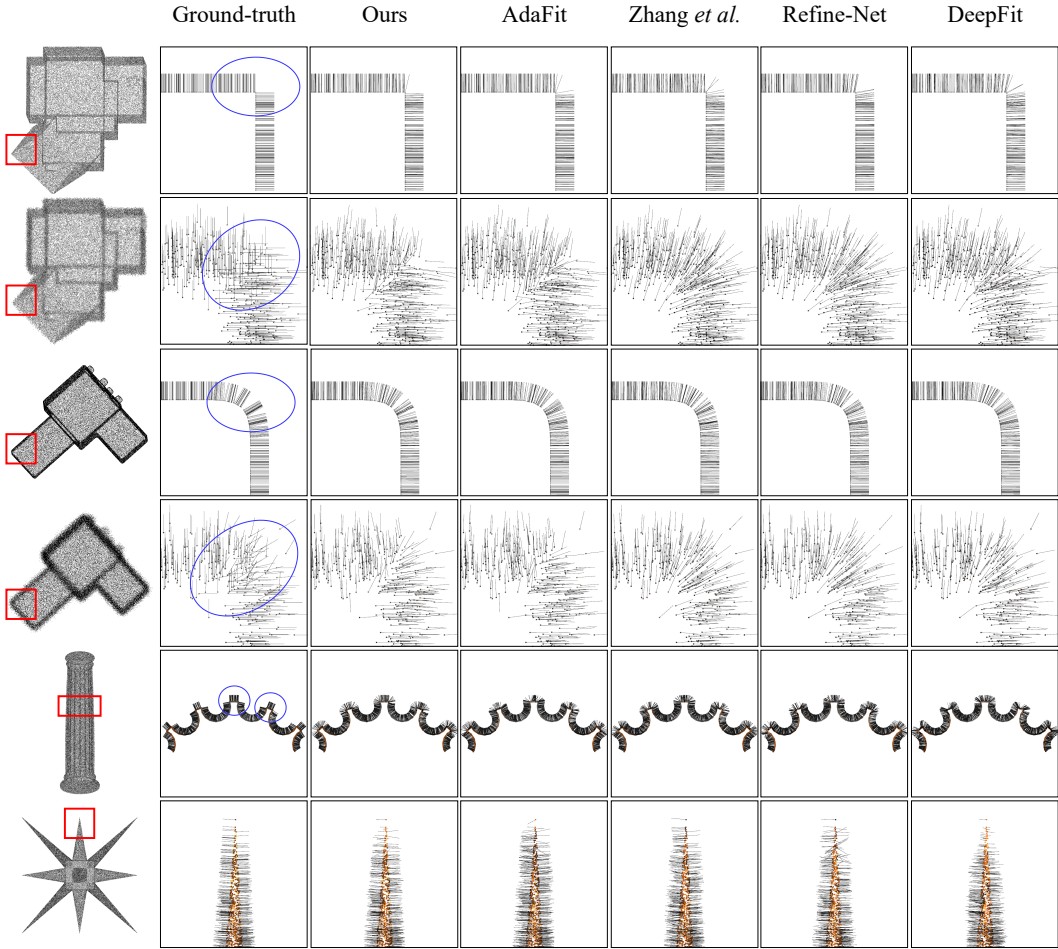

Figure 2: Cutting plane visualization of the estimated normals (black lines) in different regions with complex geometric structures.

## 3 More Experiments

### 3.1 Point Weight Visualization

In our algorithm, we use the hyper surface fitting results to predict the query point normal using an output module, in which we quantify each point's contribution to the final result by computing a weight $w_i = \text{sigmoid}(\Psi(\mathcal{N}_{\theta,\tau}(G_i, C_i)))$. In Fig. 1, we visualize how much contribution that each point in a point cloud patch makes to estimate the normal of the query. We can see that the neighbors of the query contribute more than the points that are far away from the query.

### 3.2 Additional Visualization Results

In Fig. 2, we provide the visualization results of the predicted point normal vectors in some regions with complex geometric structures, such as sharp edges and corners. Our method performs well in these regions and gives point normal vectors that are the most similar to the ground-truth normals.

In Fig. 6, we show the point clouds of the SceneNN dataset processed for normal estimation evaluation, and we also show our estimated point normals and our normal angular error maps on these point clouds. In Fig. 5, we show the point cloud of Semantic3D dataset processed for normal estimation evaluation, and we also show our estimated point normals on this point cloud. The ground-truth normal of this dataset is not available. Fig. 7 visualizes our normal estimation results on different shapes of the PCPNet dataset, where the point clouds are rendered by the estimated normal vectors.

Table 1: Comparison of the network parameter and efficiency between different learning-based normal estimation methods on the PCPNet dataset. We report the average execution time for a point cloud with 100k points.

|  | Ours | AdaFit [16] | Zhang *et al*. [11] | Refine-Net [12] | DeepFit [1] |
| --- | --- | --- | --- | --- | --- |
| Parameter (million) | 2.16 | 4.87 | 3.55 | 1.23 | 3.53 |
| Time (s) | 72.47 | 56.23 | 36.27 | - | 36.24 |

Fig. 8 shows the normal angle errors compared with the ground-truth normals on different shapes of the PCPNet dataset, where the point clouds are rendered with an error map.

## 3.3 Complexity & Efficiency

We compare our method against the related learning-based approaches that have similar normal estimation performance on the PCPNet dataset. We use the original implementation of each method and take the same point cloud data as the input. We report the number of the learnable network parameters of each model and its execution time on a single NVIDIA 2080 Ti GPU. As shown in Table 1, the parameter number of our method is relatively small. The execution time of our method has no advantage over other baselines since the local coordinate frame construction with kNN searching in our Space Transformation module takes lots of time. Compared with AdaFit [16], which has the most similar performance to our method, its network parameters are twice as much as ours. Our method formulates the surface fitting as learning hyper surfaces by MLP layers, while AdaFit approximates the local structures using the classic polynomial surface fitting. The time of Refine-Net [12] is not reported as it has a standalone data processing step, including initial normal computation, clustering, filtering, and height-map feature computation. This step is very time-consuming, and we do not count it into the execution time for a fair comparison. Zhang *et al*. [11] and DeepFit [1] take a similar execution time as they have very similar network architectures.

## 3.4 Applications

In this section, we further demonstrate typical applications using point cloud normal estimation, including point cloud filtering and rendering, which benefit directly from the accurately estimated point cloud normals. In addition, we also investigate the effectiveness of our method and baseline methods in these applications.

**Point Cloud Filtering**. In this application, we apply the method proposed in [10] to perform point cloud filtering using the point normals estimated by different methods. In Fig. 3, we visualize the qualitative comparison of the filtered point clouds and their corresponding reconstructed surfaces by Poisson algorithm. We can observe that the point cloud filtering algorithm can benefit from the normal estimated by our method, which smooths the surfaces in flat areas while still keeping detailed structures at sharp edges.

**3D Rendering/Shading**. In 3D computer graphics, the normal is often necessary for lighting computation in rendering. We use the default rendering engine in [15] to show the appearance of object surfaces with normals estimated by our method. We do not provide the visual comparison with other baseline methods as the visualization results of different methods are very similar and it is difficult to see obvious improvements.

## 4 Limitation & Broader Impact

Similar to other point cloud normal estimation methods, the performance of our method degrades substantially when dealing with exceptionally severe noise, as shown in our experiment. With the increase of noise and outliers, the underlying geometric structure of the object may be destroyed and result in many ambiguous feature points (see Fig. 2). Moreover, existing learning-based methods all need the ground-truth normal to supervise the training procedure, but it is difficult to obtain the ground-truth normal, especially for real scanned point cloud data. Unlike the object detection and segmentation benchmarks, which can get the annotations by manually labeling, it is almost impossible to label the 3D normal vectors of a point cloud with manual methods. A practical and effective solution (as most of the existing learning-based methods adopt) is to train the model on the

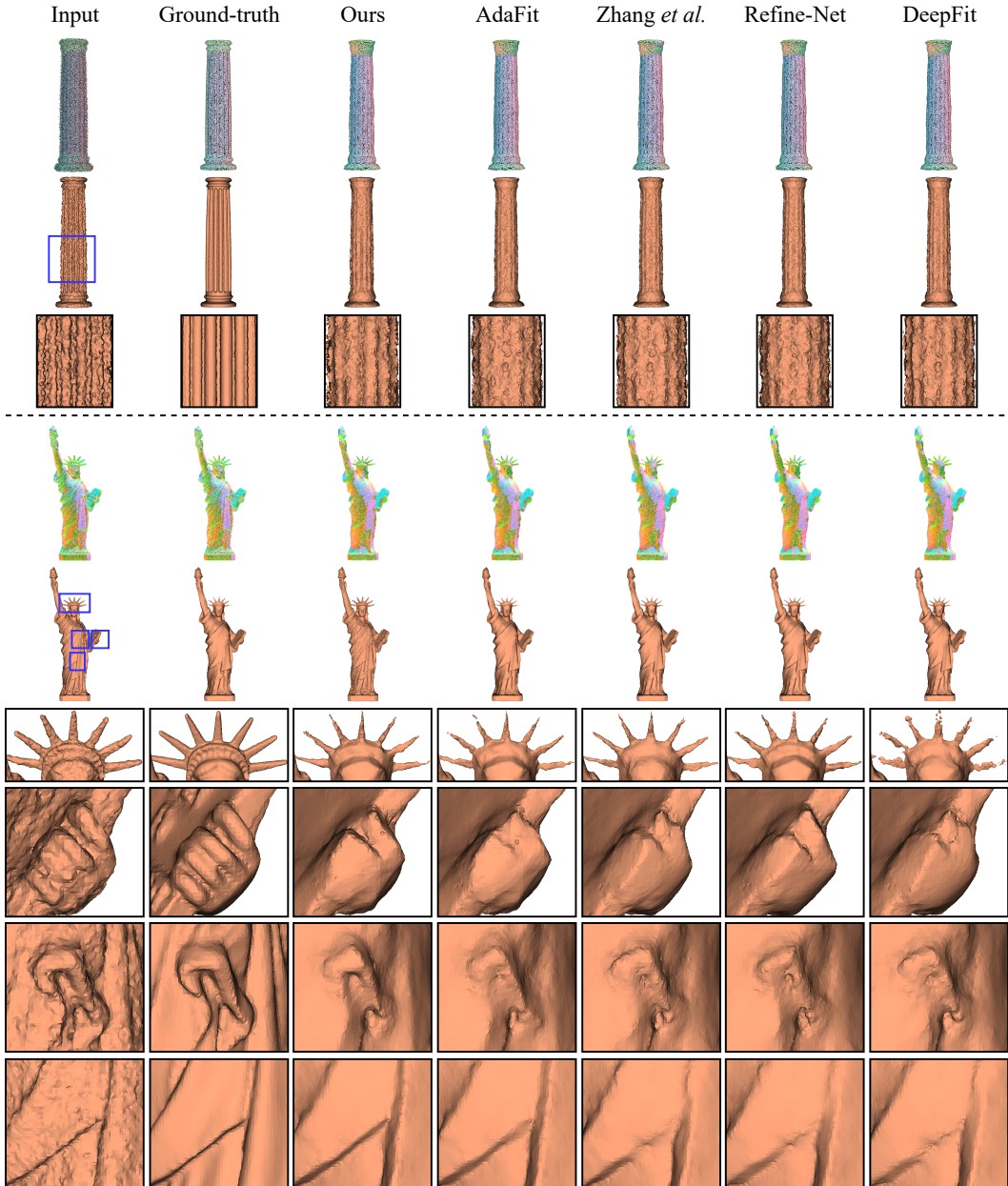

Figure 3: Qualitative comparison results of point cloud filtering using the estimated normals of different methods. The input and filtered point clouds are shown with RGB colors, and their corresponding reconstructed shape models are provided under each point cloud. The local enlarged views are provided under each shape.

synthetic dataset, where the normal is computed from the surface mesh data, and then generalize the trained model to real-world data. Finally, our method extracts local-aware features for each point, which is not efficient enough and is a major factor to limit speed. We may solve these issues by introducing novel techniques in future work.

In this work, we present a novel normal estimation method from 3D point clouds with noise, outliers and density variations. We convert the explicit polynomial surface fitting to implicitly learn hyper surfaces. For the broader academic impact, our method can benefit various downstream 3D computer vision tasks, such as point cloud filtering, surface reconstruction and rendering. The more accurate point normals will definitely enhance the performance of these algorithms, especially when dealing with complex geometric structures (see Fig. 2). Our method can also be applied to describe the object

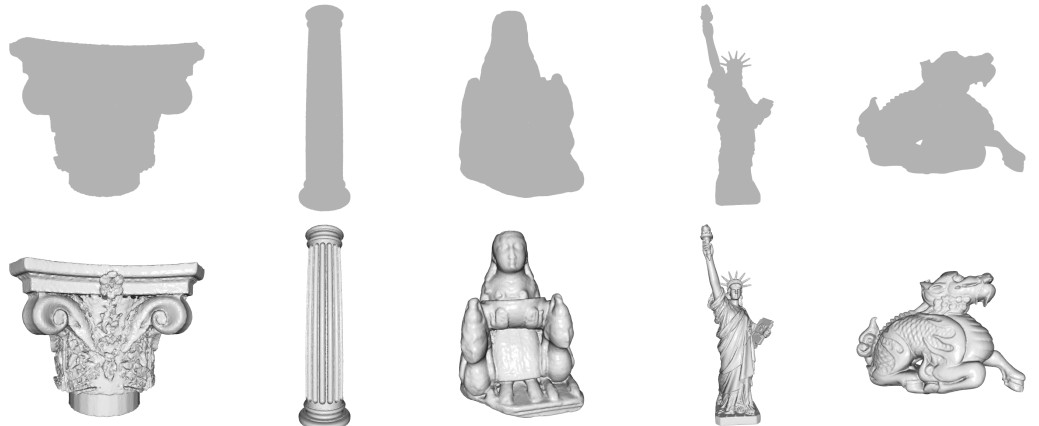

Figure 4: Top row: shape model visualization without point normals. Bottom row: rendering/shading with our estimated normals.

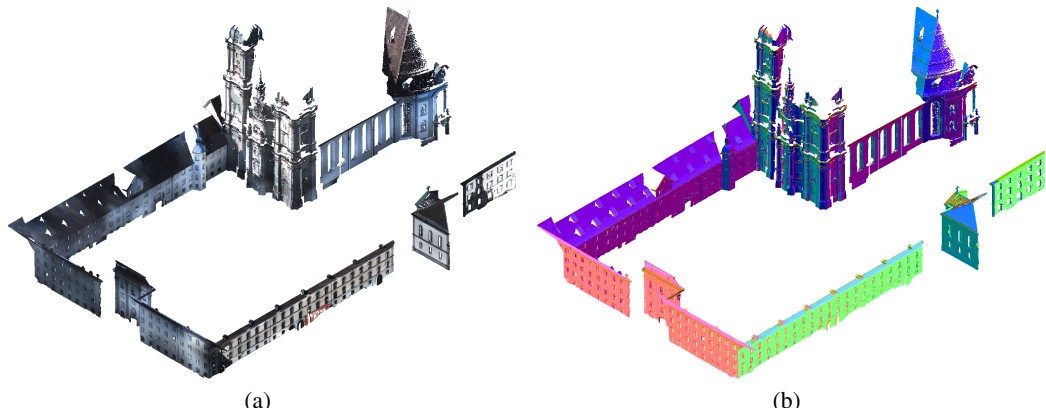

(a)          (b)

Figure 5: Visualization of the Semantic3D dataset [7] and our estimated point normals. (a) The raw 3D point cloud with RGB colors of the real-world outdoor scene. (b) Our estimated point normal vectors mapped to RGB.

surfaces for robot navigation and grasping. The excellent generation ability of our method lets the model trained on the synthetic dataset process the real-world data. The potential negative impact is that the autonomous platform may cause damage if our estimated normal is not accurate enough to make the controller identify the objects under certain special cases.

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

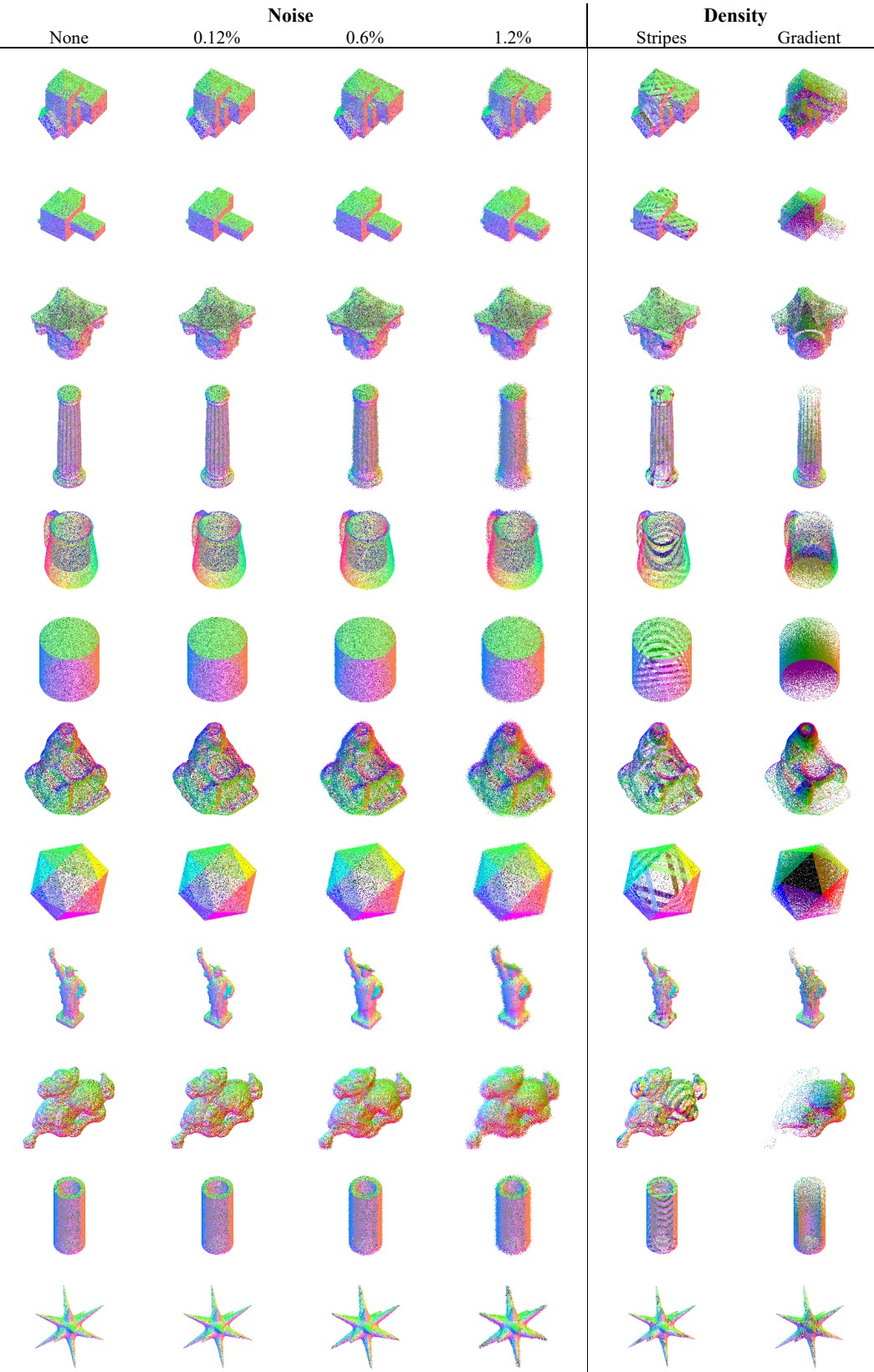

Figure 7: Ours normal estimation results for different noise levels and density variations of the PCPNet dataset [6]. The point normal vectors are mapped to RGB.

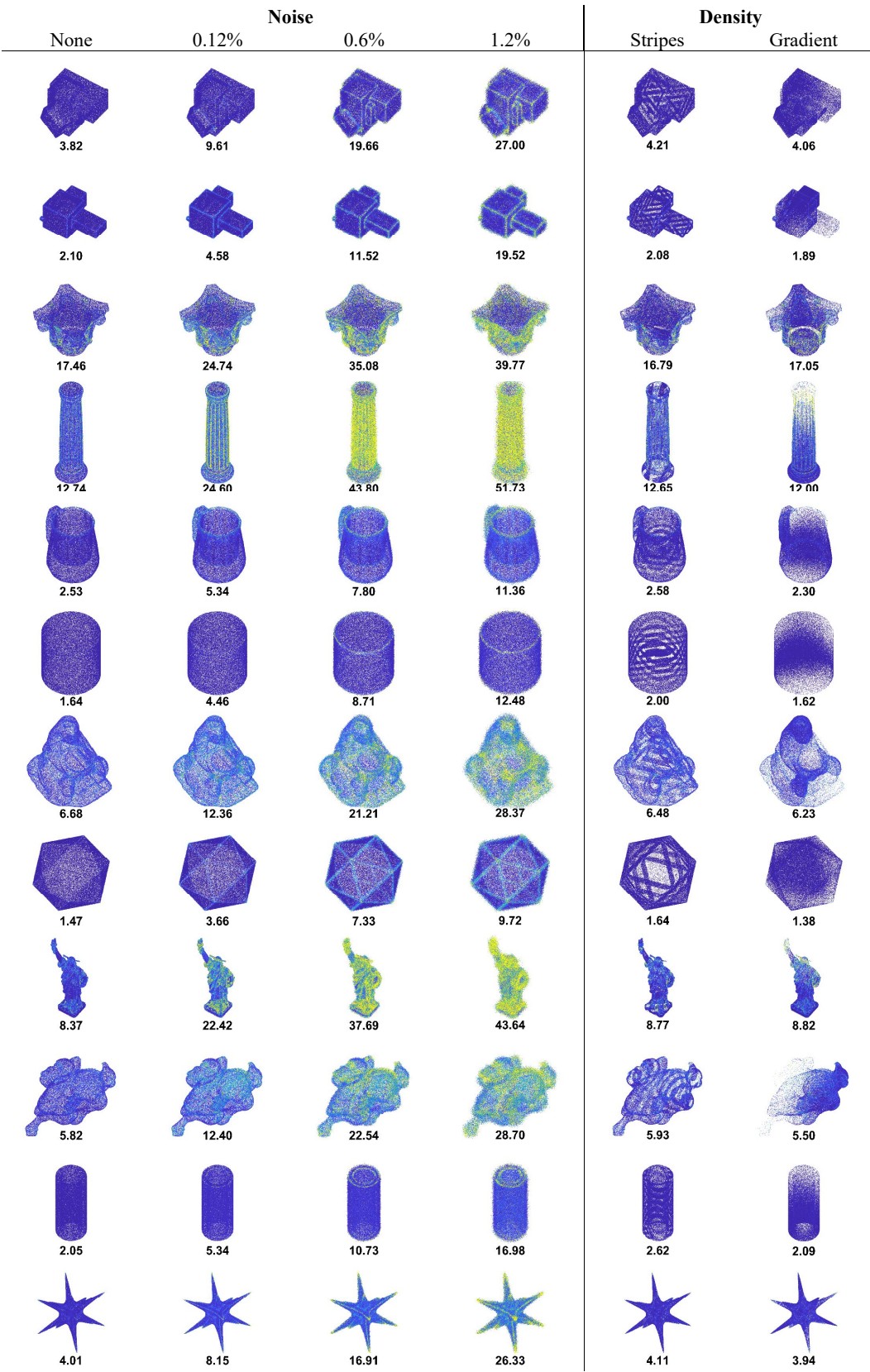

Figure 8: Error visualization for different noise levels and density variations of the PCPNet dataset [6]. The point color is the angular error mapped to a heatmap ranging from 0-40 degrees.