# OpenReview forum: "HSurf-Net: Normal Estimation for 3D Point Clouds by Learning Hyper Surfaces"
_NeurIPS.cc/2022/Conference — NeurIPS 2022 Accept_

### Official Review · Reviewer_Md6E · 2022-07-04

**Rating:** 6
**Confidence:** 5
**Soundness:** 3 good
**Presentation:** 3 good
**Contribution:** 3 good

**Summary:**

This paper proposes a new method for the task of normal estimation for point clouds using hyper-surface fitting. The model consists of relative positional encoding, and a space transformation module (MLP) to map 3D point clouds into feature space where the hyper-surfaces are fitted. The method is evaluated on three different datasets, including PCPNet, ScenNN, and Semantic 3D and shows state-of-the-art performance in RMSE and PGP metrics.

**Questions:**

There are several aspects that I am not clear on:
1. The space transformation module is unclear to me and I found Figure 3 confusing. How is it different from a PointNet++ idea? While clearly, the architecture is different the concept of multi-scale and grouping was first proposed there, so how is this different?
2. Novelty: To my understanding, this method could be interpreted as a newer version of PCPNet with PointNet++ as backbone architecture and Positional encoding as input. Each has already been proposed in previous works and showed its effectiveness and the question is how is combining these building blocks novel?
3. Insight: The main paper lacks insight into what the network is actually learning. There is some of it in the supplemental but not sure that it is enough. It would be interesting to visualize each point's contribution to the hypersurface fitting as a heatmap.
4. Will the code be released ?


**Limitations:**

The limitations are not clearly presented. Clearly, one limitation is robustness to noise. What are other limitations? When does it fail in the clean point cloud cases? Additionally, no potential societal impact was mentioned (since it is a normal estimation method, I am not sure there are any but even if there aren't, it should at least be mentioned).

**Strengths And Weaknesses:**

The paper is mostly well written and easy to follow. The method is useful as shown by the results on multiple datasets. Different elements of the model are well analysed by an ablation study and show the contribution of each of the components. Figures 1 and 2 help the reader follow along and clarify the overall framework. The supplemental material also contains interesting results.

---

> ### Author Response · Authors · 2022-08-02
> **Response to reviewer Md6E**
>
> We would like to thank the reviewer for the insightful comments.
>
> **Q1: The difference between the Space Transformation module and PointNet++.**
>
> A1: Our Space Transformation module (Fig. 3) is different from PointNet++. The way our method extracts features is different from that of PointNet++. Specifically, in order to extract the features for learning hyper surface and further estimating point normal without explicitly fitting polynomial surface, we design Local Aggregation Layer and Global Shift Layer to realize point set abstraction in our Space Transformation module, rather than directly using the PointNet++. The main differences are as follows:
>
> (a) In the Local Aggregation Layer, we group the local neighborhood features at each point by the 3D spatial distance based kNN search. Then we refine each grouped feature via a chain of Dense Block units, rather than using PointNet in PointNet++. In addition, there is no point cloud sampling during this process, so the number of points keeps unchanged.
>
> (b) In the Global Shift Layer, we provide each point with global information by fusing global features extracted from multiple neighborhood size scales of the query point p. To get the global feature, a maxpooling operation is successively performed over neighboring points of the query point p with scale size $N_s=\lbrace N, N/2, N/4\rbrace$, where $N$ is the number of points in the input patch. Because our method only estimates the query point (i.e., center point) normal in a patch, we simply select the $N_s$ nearest neighbors of the query point as a subset rather than using the Farthest Point Sampling in PointNet++.
>
> **Q2: Novelty: a newer version of PCPNet with PointNet++ and positional encoding.**
>
> A2: (a) As described in the answer to the previous question, our backbone architecture (Space Transformation module) is not PointNet++. Our Relative Positional Encoding is completely different from the traditional positional encoding which is based on sine and cosine functions. The previous positional encoding encodes the spatial position information of each point according to its coordinate. We design a parameterized and learnable encoding scheme to encode the relative position information, which reveals the local geometric structure of the point cloud. In Table 2(c) of the paper, we showed better results than the traditional position encoding. Thus, these two modules in the proposed method are novel.
>
> (b) Different from PCPNet, the most important contribution of this paper is that we propose a novel Hyper Surface representation in a high dimensional feature space and design a Hyper Surface Fitting module to optimize this surface representation for point cloud normal estimation. We analyze the problems existing in traditional polynomial surface fitting methods. We use the learned hyper surface to get the point normal in a direct regression way, instead of explicitly constructing the geometry surface. This allows our method to effectively avoid the problems caused by polynomial fitting in the normal estimation task, which achieves a significant improvement over the state-of-the-art on several datasets. Extensive ablation experiments also validate the effect of each component that contributes to the performance.
>
> **Q3: Insight: visualize each point's contribution to the hyper surface fitting.**
>
> A3: As you suggested, we have visualized and discussed each point's contribution to the hyper surface fitting in the appendix section of the revised paper.
>
> **Q4: Will the code be released?**
>
> A4: Yes, the source code and processed data will be released after acceptance.
>
> **Q5: No limitation and potential societal impact. Failure case on the clean point cloud.**
>
> A5: In Sec. 3 in the supplementary material, we have already elaborated on the limitation and broader impact of the proposed approach. In all our experiments, except for preserving some sharp corners (see Fig. 1 in the supplementary material), our method can obtain reasonable results on the clean point cloud without failure.

---

> > ### Author Response · Authors · 2022-08-08
> > **Discussion**
> >
> > Dear Reviewer Md6E,
> >
> > Following your questions, we provided additional explanations about the novelty of our approach and the difference between the Space Transformation module and PointNet++.
> > In light of this, we would like to know whether you believe we have addressed your concerns.
> >
> > Thank you for your time,
> >
> > The Authors

---

> > > ### Comment · Reviewer_Md6E · 2022-08-09
> > > **Discussion**
> > >
> > > 1. Regarding the similarity to PointNet and PointNet++ - the differences are still not highlighted enough for me. If it is an MLP operating on KNN points and aggregated over several scales this is very similar to PN++ (they do it on global shapes and here on local patches centred around the query point which PCPNet did with PN and not PN++).
> > > 2. You have clarified the point on positional encoding and addressed my concerns.
> > > 3. If the code is to be released, I expect the link to be provided in the final version of the paper.
> > > 4. In my opinion, limitations should be a part of the main paper.

---

> > > > ### Author Response · Authors · 2022-08-09
> > > > **Discussion: response to reviewer Md6E**
> > > >
> > > > Thank you very much for your reply and comments. We will update the exposition in the revised version based on your comments.
> > > >
> > > > **Q1: The similarity to PointNet and PointNet++.**
> > > >
> > > > A1:
> > > > As the feature encoder of our method, the Space Transformation module takes a 3D point cloud patch $P=\lbrace p_i|i=1,...,N \rbrace$ as the input and outputs $N/4$ point features, where the patch $P$ is centralized at the query point $p$ and the $N/4$ points are the nearest neighborhoods of the query point.
> > > > For the patch $P$, we first extract the per-point feature using a chain of Dense Block [1] and maxpooling based on the local coordinate frame of each point $p_i$.
> > > > At this stage, we do not adopt the PointNet architecture as in PCPNet.
> > > > Then we sequentially extract features from different scale neighborhoods of the query point $p$.
> > > > At this stage, we do not use the commonly used auto-encoder architecture with skip-connection and the point set abstraction in PointNet++.
> > > > Furthermore, we also do not use the farthest point sampling, the per-point kNN searching and grouping operation in PointNet++.
> > > > We get the point subsets with different scales from the input point cloud by consecutively selecting the $\lbrace N, N/2, N/4\rbrace$ neighborhoods of query point $p$.
> > > > This step is very efficient as the neighborhoods of $p$ are stored in the array with increasing distance.
> > > > We simply use the MLP and maxpooling to elevate the per-point feature dimension at scale $N$ and pass its per-point feature and global feature to the next smaller scale $N/2$, and so on.
> > > > Finally, the $N/4$ point features are fed to the following hyper surface fitting module.
> > > > As described above, our Space Transformation module is different from PointNet and PointNet++.
> > > >
> > > > [1] Huang G, Liu Z, Van Der Maaten L, et al. Densely connected convolutional networks. CVPR 2017.
> > > >
> > > > **Q2: The link to the source code.**
> > > >
> > > > A2:
> > > > As you suggested, we will provide a link to the source code in the final version of the paper.
> > > >
> > > >
> > > > **Q3: The limitation should be a part of the main paper.**
> > > >
> > > > A3:
> > > > We plan to shorten or change Sections 3.1 and 3.2 to make some space for discussion of limitations.

---

### Official Review · Reviewer_293b · 2022-07-10

**Rating:** 5
**Confidence:** 4
**Soundness:** 3 good
**Presentation:** 1 poor
**Contribution:** 2 fair

**Summary:**

The authors propose a method to estimate normals in point clouds based on a novel patch-based network architecture that integrates global patch information and local neighborhood information inside the patch in multiple steps, making heavy use of densenets and skip connections. A new cross-product-based loss is used to train the network.

I would consider the novel architecture, as well as the novel loss (which seems to be very relevant to achieve good performance according to the ablation) to be the main contributions.

**Questions:**

The authors should clearly state in the rebuttal if they plan to change the motivation of their method as described above, or alternatively if I missed something that justifies the current motivation given by the authors.

**Limitations:**

The authors have sufficiently addressed limitations and societal impact.

**Strengths And Weaknesses:**

Strengths:
- Normal estimation should be of great interest to the community.
- The architecture is reasonably novel. It roughly follows a PointNet-like setup in most components, but the component combination into the full architecture seems novel.
- The evaluation shows clear improvements over the state-of-the-art on multiple datasets, including real-world data.
- There is a thorough ablation of the design choices (even if their theoretical motivation provided by the authors is misleading, this provides some empirical motivation).
- The authors include source code and promise to provide their exact data.

Weaknesses:
- The motivation and presentation of the method is misleading.
- The exposition is missing some details of the method.
- Given that the current motivation is misleading, this leaves the design choices for the architecture poorly motivated.

Overall, due to the promising that show a clear improvement over the state-of-the-art on several datasets, I am leaning towards the positive side. However, the misleading exposition and missing details lower the quality of the paper significantly and should be corrected before the paper can be accepted. I believe that this can be done in a minor revision.

Details:
- The motivation and presentation of the method is misleading. In the introduction, Section 3.1 and Section 3.2, the approach is carefully motivated as surface fitting in a feature space. However it seems like surface fitting is not done explicitly, and instead a generic network is used that may or may not learn to do something similar as the polynomial surface fitting described in Eq. 5. The surface mathcal{N} is never explicitly constructed or encoded in any specific way in the network structure. Considering this I would clearly count the proposed method as regression-based method in the author's terminology, not as a hybrid or fitting-based method. The motivation of the approach as surface fitting seems to me like carefully motivating a method as specifically using a Sobel Filter to detect edges, and then training a generic CNN to obtain the output. It seems quite misleading and should be changed.

- The exposition is missing some details and could be improved.
  - In Section 3.1, the use of two Taylor expansions seems unnecessarily complicated. Also, since both are defined with the same order it does not become clear that the fitted surface can usually only be an approximation of the true surface. It might be clearer to define the ground truth surface with a generic function f(x,y) instead of the truncated Taylor expansion J_beta,n, and then fit J_alpha,n to this generic function.
  - The introduction should clearly mention that the goal is to find *unoriented* normals, i.e. without giving information which side of the surface is inside or outside.
  - In Eq. 5, the dimensionality of the feature space of F is not defined (the output space of mathcal{N}_theta,tau and mathcal{F})
  - The text below Eq. 5 mentions basis vectors that for G and C that are introduced in later sections, but I could not find mention of them in the rest of the text.
  - Eq. 6 is missing a vector norm for the difference between feature vectors.
  - Also, in Eq. 6, would optimizing over tau not always result in the highest possible tau being used? Since higher-order should always be able to achieve a lower error. (This may not be very relevant because no optimization is done in practice - see concerns about the motivation of the method.)
  - Section 3.2 should clarify how weights w_i are computed.
  - Losses should be discussed in the main paper, since they are non-standard and seem to be an essential part of the contribution. Shortening or changing Sections 3.1 and 3.2 could make space.
  - For Eq. 9, some motivation should be given for the design choice of using the max over all features, since this breaks the analogy to 3D surface fitting, where it would not make sense to take the max over all z values of the fitted surface.
  - In Section 3.3, it should be clarified if the kNN search for neighbors is performed in the input 3D space or the feature space. Also in Section 3.3, it should be clarified if the dense blocks are applied per-point or over the concatenated features of all grouped points in a neighborhood. If it is per-point, the local aggregation layer could be described more succinctly as a PointNet over the local neighborhood, with dense blocks instead of MLPs.
  - Section 3.3 is missing a description how G_s,i is converted to G_i. Is the last (coarsest) scale used? This should be clarified.
  - Section 3.3 and 3.4 are missing descriptions how the coarser subset of M points is chosen from the input N points. Farthest point sampling like in PointNet++? A random subset? This should be clarified.
  - In Section 3.4, the position encoding function phi is not defined. Is it an MLP as well? This should be clarified.
  - Section 3.4 is missing a description of how C^j_i is converted to C_i. Maxpooling? This should be clarified.
  - Section 4 should briefly mention how many point clouds were used from each dataset and the average point cloud size.
  - Section 4 should clarify if all baselines were trained on the same dataset.

- In the ablation, how was the MSE loss applied to the unoriented normals? This should be clarified (a naive application to output normals would not work since the normals are not oriented).

- The following papers could be added to the related work:
  - Stable and efficient differential estimators on oriented point clouds, Lejemble et al., SGP 2021
  - PCT: Point Cloud Transformer, Guo et al., CVM 2021
  - Point Transformer, Zhao et al., CVPR 2021 (they don't demonstrate normal estimation, but their architecture is for general point cloud processing)

---

> ### Author Response · Authors · 2022-08-02
> **Response to reviewer 293b**
>
> We thank the reviewer for the detailed comments. We have revised the paper based on your comments and some other details are not listed here. We will make the code publicly available to help readers understand the details of the algorithm more clearly.
>
> **Q1: The motivation and presentation are misleading.**
>
> A1: Yes, our proposed method is a regression-based method and the surface is not fitted explicitly. Eq.(5) is designed as a sequence of skip-connected Residual Blocks in the Hyper Surface Fitting module. The hyper surface $\mathcal{N}$ is embed in the network of this module, and its output $F=\mathcal{N}(G,C)$, $F=\lbrace f_1,f_2,…,f_M\rbrace$ (see Fig.2).
> As claimed in our paper, our method estimates point cloud normal by implicitly learning hyper surface rather than explicitly fitting polynomial surface. Our hyper surface is represented by MLP layers and the learnable parameters of the layers interpret the surface structure in a high dimensional feature space. The advantage is that the hyper surface can adaptively fit more complex point patterns in a robust way. The process of hyper surface fitting is equivalent to learning to determine the network parameters.
> We clearly describe the problems existing in traditional polynomial surface fitting methods in Lines 39-48, which lead to our motivation. Our method deeply draws on the idea of the traditional polynomial surface fitting. However, the biggest difference is that our method extends the 3D geometry-based surface fitting into a high dimensional feature space. We follow the polynomial surface representation (Sec.3.1) to design and describe our formulas, modules and network structures (Sec.3.2). We use the learned hyper surface to predict the point normal in a direct regression way, instead of explicitly constructing the 3D geometric surface. This allows our method to effectively avoid the problems caused by polynomial fitting, and achieve a significant improvement over the state-of-the-arts.
>
> **Q2-1: G and C are not introduced.**
>
> A2-1: G and C are introduced in Sec.3.3 and Sec.3.4 of the paper, respectively. See Eq.(10) and Eq.(11).
>
> **Q2-2: The $\tau$ in Eq.(6).**
>
> A2-2: Since we no longer use a polynomial function to fit the surface, but replace this process with a network to learn the hyper surface. Here, the $\tau$ is the parameters of an MLP-based module, which is designed as a sequence of skip-connected Residual Block units (Fig.2). It is optimized with the training of the network.
>
> **Q2-3: Eq.(9) uses the max over all features.**
>
> A2-3: Our algorithm uses Eq.(9) to estimate the normal of the query point p (i.e. center point). To estimate the normal of each point in a patch, the formula is $\mathbf{n}\_i=\mathcal{H}(\mathcal{N}_{\theta,\tau}(G_i,C_i))$ without maxpooling. In the ablation, we verified that solving the neighbor point normal of p at the same time is not helpful. In addition, the loss function only constrains the normal of the query point, so the features of the query point should be the most valuable and prominent. These motivate us to use maxpooling over all features in a patch for estimating the normal of p.
>
> **Q2-4: The kNN search and dense blocks in Sec.3.3.**
>
> A2-4: In Sec.3.3, we introduce the local aggregation layer and global shift layer. As we claimed in Line 190, in the local aggregation layer, we group the local neighborhood features at each point by the 3D spatial distance based kNN search and refine each grouped feature via a chain of Dense Block units. That means we perform the kNN search for neighbors in the input 3D space and the Dense Blocks are applied over the concatenated features of all grouped point features in a neighborhood. So, our local aggregation layer cannot be described as a PointNet over the local neighborhood. In the global shift layer, we do not use kNN search.
>
> **Q2-5: Use the last scale $G_{s,i}$ as $G_i$?**
>
> A2-5: Yes, we use the feature of the last scale $G_{s,i}$ as the $G_i$. After recursively using Eq.(10), the last output contains the information from previous scales.
>
> **Q2-6: How to choose the subset of M points?**
>
> A2-6: Because our method only solves the query point normal in a patch, we choose M points as the M nearest neighbors of the query point. In addition, the value of M is consistent with the number of points output by the Global Shift Layer.
>
> **Q2-7: How to convert $C^j_i$ to $C_i$?**
>
> A2-7: We do not define $C_i$ and there will be no $C_i$. The scheme we adopt in Hyper Surface Fitting module (Fig.2) is $F_i= MAX\lbrace MLP(C^j_i,G_i)\rbrace$. That means $C_i$ does not exist in our algorithm.
>
> **Q3: MSE loss for the unoriented normal.**
>
> A3: We use the ground-truth normal vector to make the predicted normal vector in the same direction, and then calculate the MSE loss.
>
> **Q4: Paper citations.**
>
> A4: We have added citations to these three papers in the revised version.
>
> **Q5: Change the motivation.**
>
> A5: We will make some adjustments in the final version.

---

> > ### Comment · Reviewer_293b · 2022-08-05
> > **Discussion**
> >
> > Thanks for the clarifications, I will also try to clarify some of the points I mentioned in my review, since I feel that the authors misunderstood some of them:
> >
> > **Misleading presentation**:
> > Just to clarify what I mean with the misleading presentation: I agree that the exposition claims that a hypersurface in high-dimensional space is being fitted, rather than a surface in 3D space. However, from my current understanding, I don't think that the polynomial hypersurface fitting described in Eq. 5 is necessarily what the MLP is approximating. It could also be approximating something completely different. Eq. 5 has terms like G^{k-j} C^{k} in the definition of \mathcal{N}. Are these terms explicitly created anywhere in the architecture? Please correct me if I am wrong, but I do not see any powers of G or C being formed or their products. If the argument is that the MLP is used to approximate them, this may or may not be true - there does not seem to be any explicit supervision for the MLP to approximate these terms (again, please correct me if I am wrong). Another option to convince me that these terms are formed somewhere is to show it empirically - for example that some intermediate layers of the MLP empirically approximate these terms.
> >
> > However, if it can't be shown that terms like G^{k-j} C^{k} are formed somewhere in the network, then Eq. 5 is a bit misleading and I would remove it as an interpretation of what the network is approximating. It might be approximating this, but it might also be approximating something completely different. It would be good if the authors reply what specifically they plan to change in the paper to address this (or alternatively explain what I got wrong about this).
> >
> > **Definition of G and C**:
> > I was referring to the definition of the *basis vectors* that are mentioned in the text below Eq. 5, not the definition of G and C itself. It might just require a re-formulation or a short mention somewhere in the text to clarify what is meant by these basis vectors.
> >
> > **Max-pooling in Eq. 9**
> > My main concern here is that all steps in Section 3.2 are motivated as approximating hypersurface fitting, however Eq. 9 is not motivated by hypersurface fitting (since the analogy between what the network is doing and hypersurface fitting breaks here), so a different motivation for the max-pooling needs to be given in the text. Why is specifically max-pooling used over what is supposedly the 'height' of the hypersurface to define the normal? This should be discussed in the text.

---

> > > ### Author Response · Authors · 2022-08-07
> > > **Discussion: response to reviewer 293b**
> > >
> > > Thank you very much for your reply and comments. We will update the exposition in the revised version based on your comments.
> > >
> > > **Q1: Misleading presentation.**
> > >
> > > A1: For Eq.(5) in the paper, we provide a more detailed derivation process to clarify its relationship to the designed networks.
> > > In order to expand the polynomial surface fitting in 3D dimensional space into the high dimensional feature space using a neural network with parameter $\Theta$, we define the multiplication of real numbers with order $g^\omega \odot c^\upsilon$ in the polynomial function as $\mathbf{g} \boxdot \mathbf{c}$, i.e., $g^\omega \odot c^\upsilon:=\mathbf{g} \boxdot \mathbf{c}$, and the orders $\omega,\upsilon \in [0,1,...,\tau]$. For the symbol $\boxdot$, we experimentally choose the commonly used feature concat operation (corresponding experiments and discussions will be added in the supplementary materials), hence
> > > $$
> > > \begin{aligned}
> > >     \mathcal{N}\_{\theta,\tau}(G,C) &= \sum_{k=0}^{\tau} \sum_{j=0}^{k} \theta_{k-j,j} ~ g^{k-j} c^j   \\\\
> > >         &= \theta_{0,0} + \theta_{1,0}g + \theta_{0,1}c + \cdots + \theta_{1,\tau-1}gc^{\tau-1} + \theta_{0,\tau}c^{\tau}  \\\\
> > >         &= [\theta_{0,0}, \theta_{1,0}, \theta_{0,1}, \cdots, \theta_{1,\tau-1}, \theta_{0,\tau}]
> > >             \begin{bmatrix}
> > >             1  \\\\
> > >             g  \\\\
> > >             c  \\\\
> > >             \vdots \\\\
> > >             gc^{\tau-1} \\\\
> > >             c^{\tau}  \\\\
> > >             \end{bmatrix}    \\\\
> > >         &= [\theta_{0,0}, \theta_{1,0}, \theta_{0,1}, \cdots, \theta_{1,\tau-1}, \theta_{0,\tau}]
> > >             \left(\begin{bmatrix}
> > >             g^0  \\\\
> > >             g^1  \\\\
> > >             g^0  \\\\
> > >             \vdots \\\\
> > >             g^1 \\\\
> > >             g^0  \\\\
> > >             \end{bmatrix}  \odot
> > >             \begin{bmatrix}
> > >             c^0  \\\\
> > >             c^0  \\\\
> > >             c^1  \\\\
> > >             \vdots \\\\
> > >             c^{\tau-1} \\\\
> > >             c^{\tau}  \\\\
> > >             \end{bmatrix} \right)  \\\\
> > >         &:= \Theta
> > >             \left(\begin{bmatrix}
> > >             \mathbf{g}_1  \\\\
> > >             \mathbf{g}_2  \\\\
> > >             \mathbf{g}_3  \\\\
> > >             \vdots \\\\
> > >             \mathbf{g}\_{\_{N\_{\tau}-1}} \\\\
> > >             \mathbf{g}\_{\_{N\_{\tau}}}  \\\\
> > >             \end{bmatrix}  \boxdot
> > >             \begin{bmatrix}
> > >             \mathbf{c}_1  \\\\
> > >             \mathbf{c}_2  \\\\
> > >             \mathbf{c}_3  \\\\
> > >             \vdots \\\\
> > >             \mathbf{c}\_{\_{N\_{\tau}-1}} \\\\
> > >             \mathbf{c}\_{\_{N\_{\tau}}}   \\\\
> > >             \end{bmatrix} \right)  \\\\
> > >         &= \Theta (G \boxdot C).
> > > \end{aligned}
> > > $$
> > >
> > > Then, the final bivariate function used in our hyper surface fitting is $\mathcal{N}_{\theta,\tau}(G,C)=\Theta (G \boxdot C)$, where $G$ and $C$ are high dimensional features of the 3D point clouds extracted by two different modules, which are introduced in Sec.3.3 and Sec.3.4 of the paper, respectively.
> > >
> > > **Q2: Definition of G and C.**
> > >
> > > A2: We have changed the original description in the text below Eq.(5) to: "where $G\in \mathbb{R}^c$ and $C \in \mathbb{R}^c$ are high dimensional features of the 3D point clouds extracted by two different modules, which are introduced in the following sections" and removed the concept of basis vectors.
> > >
> > > **Q3: Max-pooling in Eq. (9).**
> > >
> > > A3:In the traditional polynomial surface fitting, we have [1]
> > > $$
> > > J_{\alpha,n}(x,y) = \alpha_{0,0} + (\alpha_{1,0}x + \alpha_{0,1}y) + \frac{1}{2}(\alpha_{2,0}x^2 + 2\alpha_{1,1}xy + \alpha_{0,2}y^2) + \frac{1}{6}(\alpha_{3,0}x^3 + 3\alpha_{2,1}x^2y + \cdots) + \cdots  ~.
> > > $$
> > >
> > > The origin, that is the point of the fitted surface where the estimation is performed, is $(0, 0, \alpha_{0,0})$.
> > > The coefficients of the principal terms $(\alpha_{1,0}x + \alpha_{0,1}y)$ in the above polynomial equation are used to calculate the normal of the fitted polynomial surface at the origin
> > > $$
> > > \mathbf{n}\_{p} = h(\alpha) = (-\alpha_{1,0}, -\alpha_{0,1}, 1) / \sqrt{1 + \alpha_{1,0}^2 + \alpha_{0,1}^2}.
> > > $$
> > >
> > > The rest of the terms after the principal terms in the polynomial equation are not used in the estimation of the normal.
> > > Based on this, we use the max-pooling over all features from the hyper surface fitting to choose the most prominent feature.
> > > Then, we use a liner layer $\mathcal{H}$ to predict the normal of the query point $p$ (i.e. origin)
> > > $$
> > > \mathbf{n}\_p = || \mathcal{H}(\mathop{\rm MAX} \lbrace w_i ~ \mathcal{N}_{\theta,\tau}(G_i,C_i) | i=1,...,N \rbrace) ||,
> > > $$
> > >
> > > where $\mathcal{N}_{\theta,\tau}(G_i,C_i)$ means the per-point feature result obtained from the hyper surface fitting process.
> > >
> > > [1] Frédéric Cazals, Marc Pouget. Jet_fitting_3: A Generic C++ Package for Estimating the Differential Properties on Sampled Surfaces via Polynomial Fitting. ACM Transactions on Mathematical Software, 2008.

---

> > > > ### Comment · Reviewer_293b · 2022-08-09
> > > > **Discussion**
> > > >
> > > > Thanks for the answer on Eq. 5, however, I can still not see that powers or products are explicitly formed somewhere. Concatenation is not the same as multiplication. Additionally, something like $\mathbf{c}_5$ ($\mathbf{c}$ with subscript 5) is not the necessarily same as $c$ to the power of 5.
> > > >
> > > > I agree that a network could approximate the surface fitting described in Eq. 5, since networks can approximate any function (given enough capacity), but there is currently no reason to believe that it will actually approximate Eq. 5 and not another function.
> > > >
> > > > I would condition acceptance on either removing this misleading description, or on clearly stating in the text that this is the author's interpretation of what the network could do, but that there is currently no clear evidence that this is actually what the network is implementing.

---

> > > > > ### Author Response · Authors · 2022-08-09
> > > > > **Thanks for your helpful suggestion**
> > > > >
> > > > > Thanks for the comments.
> > > > >
> > > > > We will conduct more experiments to compare the performance obtained by our approximation with concatenation and the performance obtained with exact powers in our revision. If the former option produces better performance, we would follow your suggestion to make it very clear that our explanation is our interpretation of what the network could do.
> > > > >
> > > > > Best,
> > > > >
> > > > > Authors

---

> > > > > > ### Comment · Reviewer_293b · 2022-08-09
> > > > > > **Discussion**
> > > > > >
> > > > > > Additional experiments are always welcome, but just to clarify: I would condition acceptance on the changes I described above regardless of the outcome of such an experiment. That an explicit fitting with the approach described in Eq. 5 performs worse is not an indication that the network actually approximates Eq. 5. It would rather suggest the opposite: that the network has learned to do something different than Eq. 5, thus claiming that the network is approximating Eq. 5 is misleading.

---

### Official Review · Reviewer_o6yD · 2022-07-11

**Rating:** 5
**Confidence:** 4
**Soundness:** 3 good
**Presentation:** 3 good
**Contribution:** 3 good

**Summary:**

- This paper introduces an approach to surface normal estimation from the 3D point clouds. Instead of applying function fitting to the input data that may undergo overfitting or underfitting, the approach introduces hyper surface fitting to learn hyper surfaces in the high dimensional feature space implicitly. The approach learns hyper surfaces from the features and directly predicts normal vectors. The approach (HSurf-Net) is validated with the synthetic and real-world indoor and outdoor datasets.

**Questions:**

1. Figure 5 is hard to distinguish the proposed approach. Is there any way to visualize the errors more effectively? How about utilizing log-scale errors or changing the color map? Similarly, Figure 4 could be improved as well. Because many plots are overlapped, consider adjusting the min and max of PGP to show the other plots better. Consider using other line styles to distinguish the lines better.
2. Another suggestion is to apply the estimated normal to the various applications. For instance, making a mesh from a raw point cloud remains a challenging problem. How about showing that the noisy raw point clouds can be successfully reconstructed as a mesh using the proposed approach? Maybe using the Poisson reconstruction using the various estimation of surface normal?
3. Please answer the questions in the paper weakness section.


**Limitations:**

- The paper does not address the limitation of the proposed approach. The concern about the network overparameterization and the issue about point cloud densities should be clarified and stated in the limitation section.

**Strengths And Weaknesses:**

**Strengths**

1. The paper revisits the normal estimation problem and considers the problem as the hyper-surface fitting problem. Since the surface fitting happens on the latent feature domain, such implicit fitting is rather robust to the explicit approaches that are easily affected by additive noise. The reviewer is not fully tracking the literature on this field, but it seems novel and interesting.
2. The paper comprehensively summarizes the related work in the normal estimation field. The paper is self-contained, so the readers readily follow the problem and the recent advances in this field.
3. The paper is straightforward to understand, and the approach shows compelling results on the PCPNet, SceneNN, and Semantic3D datasets. The visual quality of the estimated normal is reasonable.
4. The paper explains the proposed idea in detail. The supportive figures (such as Figures 2 and 3) help to understand the approach better.

**Weakness**
1. It is unclear that the proposed approach is completely free from ‘underfitting and overfitting'. The results may depend on the ‘c’ dimension of the hyper surface coefficients (Sec. 3.2). Even if the relevant features and network output would fit for the ‘c’, there is a danger that network capacity would be overparameterized. Can the authors mention this? This is an important issue because the main argument of the proposed approach is expressed in this manner. For instance, lines 7-9 “fitting surfaces explicitly from arw point clouds suffers from overfitting or underfitting issues caused by inappropriate polynomial orders and outliers… To address these issues, we introduce hyper surface fitting to implicitly learn hyper-surfaces…”
2. The relative position encoding module directly embeds the relative position in a local frame (p^j_j-p_i). However, such embedding would work for the specific metric scale. How would the approach handle the point clouds with non-metric scale point clouds? What I mean is that the scale set of the patch size {N, N/2, N/4} would be invariant regardless of the scale of the point cloud, but (p^j_j-p_i) would be directly affected by the scale of the point cloud. In other words, if there are two point clouds {X} and {X/10}, can the network would produce the same output?
3. It is interesting to see that Eq (9) induces the normal vector. However, the equation itself does not guarantee ||n||=1 because it is the network output of the max-pooled feature vectors. Does any additional post-processing is applied to get the normalized n? Please clarify.
4. One critical concern is whether the approach would handle uneven point clouds. The paper mentions the ‘patch’ over the paper, and mention that ‘patch size’ as the number of the points (Line224 and so on). However, such a definition of ‘patches’ is readily affected by the density of the point cloud. For instance, in the KITTI dataset, in the coarse region, if we set N=700, it will introduce severely biased point cloud samples. The all experiments are conducted with the evenly sampled point clouds, where the density of the point cloud is fixed over the dataset. Please state how the proposed approach would handle density changes or uneven point clouds.
5. In the experiment section, the paper states the proposed approach outperforms the previous approaches, but it would be better if the paper mentioned the comparison between the second-best approach. For instance, AdaFit[45] is quite similar to the proposed approach. Please provide some discussion or analysis of the suggested approach.

---

> ### Author Response · Authors · 2022-08-02
> **Response to reviewer o6yD**
>
> We would like to thank the reviewer for the insightful comments.
>
> **Q1: The underfitting and overfitting. The dimension ‘c’.**
>
> A1: Generally, due to the limitations of the network itself and the complexity of the data, it is difficult to make a method absolutely free from underfitting and overfitting. Compared with existing methods, experimental results show that our method can overcome this problem to a large extent. In existing fitting based algorithms, underfitting and overfitting are mainly caused by artificially pre-determining a polynomial order in the process of polynomial surface fitting, and the order may not be suitable to the complexity of unpredictable data. Due to the limitations of human knowledge and experience, an optimal polynomial function is usually difficult to be formulated for the given data. The neural network has the advantages of adaptively learning the order and reasonably fitting the data by learning from a large amount of data. The underfitting and overfitting will not invalidate the algorithm, but it does affect the performance of the method.
> To further verify the relationship between the network capability and the dimension ‘c’, we use different dimensions to verify the performance of the algorithm on the PCPNet dataset and compare it with SOTA AdaFit [ICCV 2021]. The results are shown in the following table.
> Category|Clean|0.12%|0.6%|1.2%|Stripes|Gradient|Average
> -|-|-|-|-|-|-|-
> AdaFit|5.19|9.05|16.45|21.94|6.01|5.90|10.76
> Ours(32)|4.33|8.81|16.24|21.65|5.17|5.02|10.21
> Ours(64)|4.53|8.84|16.24|21.64|5.40|5.07|10.28
> Ours(128)|4.17|8.78|16.25|21.61|4.98|4.86|10.11
> Ours(256)|4.18|8.78|16.23|21.65|5.06|4.96|10.14
>
> Based on this experiment, we typically select 128 as the dimension that allows the algorithm to achieve the best performance, and we use it in experiments of the paper. The results also show that our method has better performance than AdaFit under all dimensions [32, 64, 128, 256].
>
> **Q2: Handle non-metric scale point clouds.**
>
> A2: As described in Line 124 of the paper, the input patch is normalized with its patch radius, and the query point is used as the origin to transform the patch into a unified coordinate system. As shown in Fig.2, our Relative Position Encoding module and the Space Transformation module are working in parallel, but only the transformation module processes the point cloud of different scales. We select $M=N/4$ neighboring points of the query point as the input of the encoding module, and the number of points keeps the same during the processing. Thus, $(p^j_i-p_i)$ in the encoding module will not be affected by different patch size scales.
>
> **Q3: About $||\mathbf{n}||=1$.**
>
> A3: We use an MLP layer to output 3D normal vectors, and then use an additional normalization to ensure that the output vector $||\mathbf{n}||=1$. We have added the corresponding symbol to indicate this step in Eq.(9) in the revised version.
>
> **Q4: Handle uneven point clouds.**
>
> A4: We have already verified the effectiveness of our method on unevenly sampled point clouds in experiments. The PCPNet dataset contains unevenly sampled data. Please see the category of density stripes and gradient in Table 1. Examples of point clouds for this dataset are shown in Fig.7 and Fig.8 in the supplementary material. The Semantic3D dataset is scanned by LiDAR in real-world outdoor scenes, and its data is also unevenly distributed, please see Fig.5 in the supplementary material. The performance of all algorithms on uneven point clouds does deteriorate, but our results are still the best.
>
> **Q5: Discussion about AdaFit.**
>
> A5: Our method is completely different from AdaFit. Both AdaFit and DeepFit adopt the PointNet to regress the weights of each point in a patch, and then use the traditional polynomial surface fitting to explicitly fit a 3D geometric surface and solve the normal vector of the surface as the point normal. On the contrary, our method implicitly learns the hyper surface to directly regress the 3D normal vector of the point without requiring any fitting for a geometric surface. We have already shown the difference between our method and AdaFit in Fig.1 of the paper and illustrated the problems of geometry-based polynomial fitting methods such as AdaFit in Lines 39-48. Moreover, all experimental results show that our method achieves better performance than AdaFit.
>
> **Q6: Figures 4 and 5.**
>
> A6: As you suggested, we will update Figures 4 and 5 in the revised version.
>
> **Q7: Lack of applications.**
>
> A7: In Sec.2.3 in the supplementary material, we have already shown three applications using the estimated normal.
>
> **Q8: No limitation.**
>
> A8: In Sec.3 in the supplementary material, we have already elaborated on the limitation and broader impact of our approach. We have verified its effectiveness on unevenly sampled point clouds in Table 1 of the paper. As you suggested, we will add a discussion about the network overparameterization in the revised version.

---

> > ### Author Response · Authors · 2022-08-08
> > **Discussion**
> >
> > Dear Reviewer o6yD,
> >
> > We analyzed the relationship between the network overparameterization and the ‘c’ dimension and discussed the difference between the proposed method and AdaFit.
> > In light of this, we would like to know whether you believe we have addressed your concerns, and if so we hope that you would be willing to increase your score.
> >
> > Thank you for your time,
> >
> > The Authors

---

### Official Review · Reviewer_KzzY · 2022-07-22

**Rating:** 7
**Confidence:** 3
**Soundness:** 3 good
**Presentation:** 3 good
**Contribution:** 3 good

**Summary:**

The authors propose a novel normal estimation method called HSurf-Net. The method works by implicitly learning hyper surfaces in a high dimensional feature space. The method includes a novel space transformation module consisting of a sequence of local aggregation layers and global shift layers, to reliably build the feature space. Experimental results show that the HSurf-Net can accurately predict normals from point clouds with noise and density variations, and achieves state-of-the-art performance on multiple datasets.

**Questions:**

- There are a lot of different notations (especially hyper-parameters) in this paper. Organizing them into a table in the appendix may help readers to follow the paper more easily.
- The idea of hierarchical point set abstraction (Fig.3) is actually proposed by PointNet++. The authors might want to cite this paper.
- Include a real-world example (e.g. a lidar scan) could further strengthen the paper.
- According to the theory proposed in this paper. The points are lifted to noise free features before fitting hyper surfaces. Could we utilize this feature to perform other point processing tasks (maybe jointly) e.g. denoising / super-resolution? Any discussions on this would be appreciated.

**Limitations:**

Yes, in the supplementary material.

**Strengths And Weaknesses:**

Strengths
- The paper is well-written and easy to follow. The method is technically sound. The authors have done abundant experiments covering a broad range of baselines and datasets, making the results convincing.
- The idea of transforming points into a higher dimensional feature space and fitting hyper surfaces is interesting and novel. The authors have designed the network accordingly to achieve this goal and performed ablation studies to validate each design choice.
- The method has shown impressive robustness towards noisy inputs. The HSurf-Net outperforms all other baselines under all settings.

Weaknesses
This paper is well-presented and proposes a novel model for normal estimation with state-of-the-art performance. I do not see any major issue in its current form.

---

> ### Author Response · Authors · 2022-08-02
> **Response to reviewer KzzY**
>
> We would like to thank the reviewer for the insightful comments, in particular for the utilization of features.
>
> **Q1: Organize notations into a table.**
>
> A1: As you suggested, we have added a table of notations in the appendix section of the revised paper.
>
> **Q2: The hierarchical point set abstraction is proposed by PointNet++. Cite the paper.**
>
> A2: We have added a reference to PointNet++ in the revised paper. In fact, our hierarchical point set abstraction is different from the one proposed by PointNet++ and our Space Transformation module (Fig. 3) is also different from the encoder of PointNet++. In order to extract the features for learning hyper surface and further predicting point normal without explicitly fitting polynomial surface, we specifically design Local Aggregation Layer and Global Shift Layer to realize point set abstraction in the Space Transformation module, rather than directly using the PointNet++. The main differences are as follows.
>
> (a) In the Local Aggregation Layer, we group the local neighborhood features at each point by the 3D spatial distance based kNN search. Then we refine each grouped feature via a chain of Dense Block units, rather than using PointNet in PointNet++. In addition, there is no point cloud sampling during this process, so the number of points keeps unchanged.
>
> (b) In the Global Shift Layer, we provide each point with global information by fusing global features extracted from multiple neighborhood size scales of the query point p. To get the global feature, a maxpooling operation is successively performed over neighboring points of the query point p with scale size $N_s=\lbrace N, N/2, N/4\rbrace$, where $N$ is the number of points in the input patch. Because our method only estimates the normal of the query point (i.e., center point) in a patch, we simply select the $N_s$ nearest neighbors of the query point as a subset rather than using farthest point sampling in PointNet++.
>
> **Q3: Include a real-world LiDAR example.**
>
> A3: Actually, both Fig. 6 in the main paper and Fig. 5 in the supplementary material show the normal results on the Semantic3D dataset (real-world outdoor LiDAR scan), and Fig. 6 in the main paper is a partial enlarged view of the Semantic3D dataset. As you suggested, we will present more results on LiDAR data in the supplementary material.
>
> **Q4: Utilize the noise-free feature to perform other point processing.**
>
> A4: We believe that such noise-free features based on normal vector estimation tasks should also be helpful for tasks such as point cloud denoising and super-resolution, and we have similar ideas and attempts. Generally, traditional methods implicitly or explicitly determine a surface when solving tasks such as normal vector estimation, point cloud denoising, surface reconstruction, and super-resolution. For example, the normal vector estimation task needs to estimate the vertical vector according to the local plane, the point cloud denoising task needs to pull noisy points onto the surface, and the surface reconstruction task needs to determine a zero iso-surface about SDF, so there is a strong correlation among these tasks. The results of the application experiments in the supplementary material show that better normal vectors enable traditional methods, such as point cloud denoising and surface reconstruction, to achieve better results.
> Our method implicitly learns hyper surfaces to estimate point normals rather than explicitly fitting polynomial surfaces, and achieve the-state-of-art performance. We find that our local feature aggregation layer is similar in design to the work ‘Score-Based Point Cloud Denoising’ [ICCV 2021]. Unlike this work, which focuses on the offset of each point, we focus on the normal vector of the query point, so we use Global Shift Layer to obtain the features of different neighborhood scales of the query point. This shows that the network structures used in solving such tasks have a certain generality.
> In conclusion, we believe that features in normal vector estimation can lead to better results for other related tasks with deep learning based schemes, but the condition is that we need to design corresponding reasonable constraints according to the specific task. In our denoising experiments with this feature, we need to add new constraints about smooth surfaces, otherwise simply using the features in the normal vector estimation task will not get good denoising results. We will continue to investigate the applicability of this feature in different tasks in the follow-up work.

---

> > ### Author Response · Authors · 2022-08-08
> > **Discussion**
> >
> > Dear Reviewer KzzY,
> >
> > We revised the paper based on your comments and provided a discussion about utilizing the noise-free feature to perform other point processing.
> > In light of this, we would like to know whether you believe we have addressed your concerns.
> >
> > Thank you for your time,
> >
> > The Authors

---

### Meta-Review · Area_Chair_ukDD · 2022-08-28

**Recommendation:** Accept
**Confidence:** Certain

**Metareview:**

This paper proposes an approach to fit implicit surfaces for surface normal estimation. Reviewers unanimously agree on its novelty and performance. AC hence recommends acceptance.

**Award:**

No

---

### Decision · Program_Chairs · 2022-09-14

Accept